# Quantitative Long-Term Monitoring of the Circulating Gases in the KATRIN Experiment Using Raman Spectroscopy

**DOI:** 10.3390/s20174827

**Published:** 2020-08-26

**Authors:** Max Aker, Konrad Altenmüller, Armen Beglarian, Jan Behrens, Anatoly Berlev, Uwe Besserer, Benedikt Bieringer, Klaus Blaum, Fabian Block, Beate Bornschein, Lutz Bornschein, Matthias Böttcher, Tim Brunst, Thomas C. Caldwell, Suren Chilingaryan, Wonqook Choi, Deseada D. Díaz Barrero, Karol Debowski, Marco Deffert, Martin Descher, Peter J. Doe, Otokar Dragoun, Guido Drexlin, Stephan Dyba, Frank Edzards, Klaus Eitel, Enrico Ellinger, Ralph Engel, Sanshiro Enomoto, Mariia Fedkevych, Arne Felden, Joseph F. Formaggio, Florian Fränkle, Gregg B. Franklin, Fabian Friedel, Alexander Fulst, Kevin Gauda, Woosik Gil, Ferenc Glück, Robin Größle, Rainer Gumbsheimer, Volker Hannen, Norman Haußmann, Klaus Helbing, Stephanie Hickford, Roman Hiller, David Hillesheimer, Dominic Hinz, Thomas Höhn, Thibaut Houdy, Anton Huber, Alexander Jansen, Christian Karl, Jonas Kellerer, Luke Kippenbrock, Manuel Klein, Christoph Köhler, Leonard Köllenberger, Andreas Kopmann, Marc Korzeczek, Alojz Kovalík, Bennet Krasch, Holger Krause, Luisa La Cascio, Thierry Lasserre, Thanh-Long Le, Ondřej Lebeda, Bjoern Lehnert, Alexey Lokhov, Moritz Machatschek, Emma Malcherek, Alexander Marsteller, Eric L. Martin, Matthias Meier, Christin Melzer, Susanne Mertens, Klaus Müller, Simon Niemes, Patrick Oelpmann, Alexander Osipowicz, Diana S. Parno, Alan W.P. Poon, Jose M. Lopez Poyato, Florian Priester, Oliver Rest, Marco Röllig, Carsten Röttele, R.G. Hamish Robertson, Caroline Rodenbeck, Milos Ryšavỳ, Rudolf Sack, Alejandro Saenz, Peter Schäfer, Anna Schaller (née Pollithy), Lutz Schimpf, Klaus Schlösser, Magnus Schlösser, Lisa Schlüter, Michael Schrank, Bruno Schulz, Michal Sefčík, Hendrik Seitz-Moskaliuk, Valérian Sibille, Daniel Siegmann, Martin Slezák, Felix Spanier, Markus Steidl, Michael Sturm, Menglei Sun, Helmut H. Telle, Larisa A. Thorne, Thomas Thümmler, Nikita Titov, Igor Tkachev, Drahoš Vénos, Kathrin Valerius, Ana P. Vizcaya Hernández, Marc Weber, Christian Weinheimer, Christiane Weiss, Stefan Welte, Jürgen Wendel, John F. Wilkerson, Joachim Wolf, Sascha Wüstling, Weiran Xu, Yung-Ruey Yen, Sergey Zadoroghny, Genrich Zeller

**Affiliations:** 1Tritium Laboratory Karlsruhe (TLK), Karlsruhe Institute of Technology (KIT), Hermann-von-Helmholtz-Platz 1, 76344 Eggenstein-Leopoldshafen, Germany; max.aker@kit.edu (M.A.); uwe.besserer@kit.edu (U.B.); beate.bornschein@kit.edu (B.B.); robin.groessle@kit.edu (R.G.); david.hillesheimer@kit.edu (D.H.); bennet.krasch@kit.edu (B.K.); thanh-long.le@kit.edu (T.-L.L.); alexander.marsteller@kit.edu (A.M.); christin.melzer@kit.edu (C.M.); simon.niemes@kit.edu (S.N.); florian.priester@kit.edu (F.P.); marco.roellig@kit.edu (M.R.); carsten.roettele@kit.edu (C.R.); peter.schaefer2@kit.edu (P.S.); michael.sturm@kit.edu (M.S.); stefan.welte@kit.edu (S.W.); juergen.wendel@kit.edu (J.W.); genrich.zeller@kit.edu (G.Z.); 2Department of Physics, Technische Universität München, James-Franck-Str. 1, 85748 Garching, Germany; konrad.altenmueller@ph.tum.de (K.A.); tbrunst@mpp.mpg.de (T.B.); edzards@mpp.mpg.de (F.E.); thoudy@mpp.mpg.de (T.H.); karlch@mpp.mpg.de (C.K.); ckoehler@mpp.mpg.de (C.K.); matthias.a.meier@tum.de (M.M.); mertens@mpp.mpg.de (S.M.); pollithy@mpp.mpg.de (A.S.); lisa.schlueter@mpp.mpg.de (L.S.); siegmann@mpp.mpg.de (D.S.); slezak@mpp.mpg.de (M.S.); 3IRFU (DPhP & APC), CEA, Université Paris-Saclay, 91191 Gif-sur-Yvette, France; thierry.lasserre@cea.fr; 4Institute for Data Processing and Electronics (IPE), Karlsruhe Institute of Technology (KIT), Hermann-von-Helmholtz-Platz 1, 76344 Eggenstein-Leopoldshafen, Germany; armen.beglarian@kit.edu (A.B.); suren.chilingaryan@kit.edu (S.C.); andreas.kopmann@kit.edu (A.K.); marc.weber@kit.edu (M.W.); sascha.wuestling@kit.edu (S.W.); 5Institute of Experimental Particle Physics (ETP), Karlsruhe Institute of Technology (KIT), Wolfgang-Gaede-Str. 1, 76131 Karlsruhe, Germany; jan.behrens@kit.edu (J.B.); fabian.block@kit.edu (F.B.); wonqook.choi@kit.edu (W.C.); marco.deffert@kit.edu (M.D.); martin.descher@kit.edu (M.D.); guido.drexlin@kit.edu (G.D.); fabian.friedel@kit.edu (F.F.); stephanie.hickford@kit.edu (S.H.); roman.hiller@kit.edu (R.H.); anton.huber@kit.edu (A.H.); jonas.kellerer@kit.edu (J.K.); manuel.klein@kit.edu (M.K.); marc.korzeczek@kit.edu (M.K.); luisa.cascio@kit.edu (L.L.C.); moritz.machatschek@student.kit.edu (M.M.); lutz.schimpf@kit.edu (L.S.); hendrik.seitz-moskaliuk@kit.edu (H.S.-M.); joachim.wolf@kit.edu (J.W.); 6Institute for Nuclear Physics (IKP), Karlsruhe Institute of Technology (KIT), Hermann-von-Helmholtz-Platz 1, 76344 Eggenstein-Leopoldshafen, Germany; lutz.bornschein@kit.edu (L.B.); klaus.eitel@kit.edu (K.E.); ralph.engel@kit.edu (R.E.); arne.felden@kit.edu (A.F.); florian.fraenkle@kit.edu (F.F.); woosik.gil@kit.edu (W.G.); ferenc.glueck@kit.edu (F.G.); rainer.gumbsheimer@kit.edu (R.G.); dominic.hinz@kit.edu (D.H.); thomas.hoehn@kit.edu (T.H.); alexander.jansen@kit.edu (A.J.); leonard.koellenberger@kit.edu (L.K.); holger.krause@kit.edu (H.K.); emma.malcherek@kit.edu (E.M.); klaus.mueller2@kit.edu (K.M.); klaus.schloesser@kit.edu (K.S.); michael.schrank@kit.edu (M.S.); felix.spanier@kit.edu (F.S.); markus.steidl@kit.edu (M.S.); thomas.thuemmler@kit.edu (T.T.); kathrin.valerius@kit.edu (K.V.); 7Institute for Nuclear Research of Russian Academy of Sciences, 60th October Anniversary Prospect 7a, 117312 Moscow, Russia; berlev@inr.ru (A.B.); titov@inr.ru (N.T.); tkachev@ms2.inr.ac.ru (I.T.); serge@inr.ru (S.Z.); 8Institut für Kernphysik, Westfälische Wilhelms-Universität Münster, Wilhelm-Klemm-Str. 9, 48149 Münster, Germany; benedikt.b@wwu.de (B.B.); m_boet07@uni-muenster.de (M.B.); st.dyba@uni-muenster.de (S.D.); kostina@uni-muenster.de (M.F.); a_fuls01@uni-muenster.de (A.F.); k.gauda@uni-muenster.de (K.G.); hannen@uni-muenster.de (V.H.); alexey.lokhov@uni-muenster.de (A.L.); patrick.oelpmann@uni-muenster.de (P.O.); o_rest01@uni-muenster.de (O.R.); rodenbeck@wwu.de (C.R.); sackr@uni-muenster.de (R.S.); weinheim@uni-muenster.de (C.W.); 9Max-Planck-Institut für Kernphysik, Saupfercheckweg 1, 69117 Heidelberg, Germany; klaus.blaum@mpi-hd.mpg.de; 10Max-Planck-Institut für Physik, Föhringer Ring 6, 80805 München, Germany; 11Department of Physics and Astronomy, University of North Carolina, Chapel Hill, NC 27599, USA; tcald@unc.edu (T.C.C.); elmartin@unc.edu (E.L.M.); jfw@unc.edu (J.F.W.); 12Triangle Universities Nuclear Laboratory, Durham, NC 27708, USA; 13Departamento de Química Física Aplicada, Universidad Autónoma de Madrid, Campus de Cantoblanco, 28049 Madrid, Spain; deseada.diaz@uam.es (D.D.D.B.); jml.poyato@uam.es (J.M.L.P.); 14Department of Physics, Faculty of Mathematics and Natural Sciences, University of Wuppertal, Gaußstr. 20, 42119 Wuppertal, Germany; debowski@uni-wuppertal.de (K.D.); ellinger@uni-wuppertal.de (E.E.); haussmann@uni-wuppertal.de (N.H.); helbing@uni-wuppertal.de (K.H.); 15Center for Experimental Nuclear Physics and Astrophysics, and Department of Physics, University of Washington, Seattle, WA 98195, USA; pdoe@uw.edu (P.J.D.); sanshiro@uw.edu (S.E.); lkippenb@gmail.com (L.K.); rghr@uw.edu (R.G.H.R.); mengleis@uw.edu (M.S.); 16Nuclear Physics Institute of the CAS, v.v.i., CZ-250 68 Husinec—Řež, Czech Republic; dragoun@ujf.cas.cz (O.D.); kovalik@jinr.ru (A.K.); lebeda@ujf.cas.cz (O.L.); rysavy@ujf.cas.cz (M.R.); sefcik@ujf.cas.cz (M.S.); venos@ujf.cas.cz (D.V.); 17Laboratory for Nuclear Science, Massachusetts Institute of Technology, 77 Massachusetts Ave, Cambridge, MA 02139, USA; josephf@mit.edu (J.F.F.); vsibille@mit.edu (V.S.); weiranxu@mit.edu (W.X.); 18Department of Physics, Carnegie Mellon University, Pittsburgh, PA 15213, USA; gbfranklin@cmu.edu (G.B.F.); dparno@cmu.edu (D.S.P.); lthorne@andrew.cmu.edu (L.A.T.); avizcaya@andrew.cmu.edu (A.P.V.H.); yyen@andrew.cmu.edu (Y.-R.Y.); 19Institute for Nuclear and Particle Astrophysics and Nuclear Science Division, Lawrence Berkeley National Laboratory, Berkeley, CA 94720, USA; bjoern.lehnert@lbl.gov (B.L.); awpoon@lbl.gov (A.W.P.P.); 20Department of Electrical Engineering and Information Technology, University of Applied Sciences (HFD) Fulda, Leipziger Str. 123, 36037 Fulda, Germany; alexander.osipowicz@et.hs-fulda.de; 21Institut für Physik, Humboldt-Universität zu Berlin, Newtonstr. 15, 12489 Berlin, Germany; saenz@physik.hu-berlin.de (A.S.); bruno@physik.hu-berlin.de (B.S.); 22Project, Process, and Quality Management (PPQ), Karlsruhe Institute of Technology (KIT), Hermann-von-Helmholtz-Platz 1, 76344 Eggenstein-Leopoldshafen, Germany; christiane.weiss@kit.edu

**Keywords:** Raman spectroscopy, tritium, gas composition monitoring, KATRIN

## Abstract

The Karlsruhe Tritium Neutrino (KATRIN) experiment aims at measuring the effective electron neutrino mass with a sensitivity of 0.2 eV/c^2^, i.e., improving on previous measurements by an order of magnitude. Neutrino mass data taking with KATRIN commenced in early 2019, and after only a few weeks of data recording, analysis of these data showed the success of KATRIN, improving on the known neutrino mass limit by a factor of about two. This success very much could be ascribed to the fact that most of the system components met, or even surpassed, the required specifications during long-term operation. Here, we report on the performance of the laser Raman (LARA) monitoring system which provides continuous high-precision information on the gas composition injected into the experiment’s windowless gaseous tritium source (WGTS), specifically on its isotopic purity of tritium—one of the key parameters required in the derivation of the electron neutrino mass. The concentrations *c*_x_ for all six hydrogen isotopologues were monitored simultaneously, with a measurement precision for individual components of the order 10^−3^ or better throughout the complete KATRIN data taking campaigns to date. From these, the tritium purity, *ε*_T_, is derived with precision of <10^−3^ and trueness of <3 × 10^−3^, being within and surpassing the actual requirements for KATRIN, respectively.

## 1. Introduction

The (electron) neutrino was postulated and theoretically described in the early 1930s to explain the conservation of energy, momentum, and angular momentum (spin) during β-decay [1,2]. However, it took about a quarter of a century, until 1956, to confirm experimentally that neutrinos exist [3]. The chargeless neutrino, ν, and its antiparticle (the antineutrino), $\bar{\nu }$, belong to the lepton family of particles which form part of the very successful Standard Model of particle physics. The Standard Model was developed in the mid-1970s, and then—based on the 20 years of experimental work since the neutrino discovery—its construct assumed that (i) corresponding to the charged leptons (e, μ, and τ), there were three neutrino flavors ($\nu_{e}$, $\nu_{\mu }$, and $\nu_{\tau }$), and their antiparticles); (ii) all neutrinos are left-handed (spin −1/2), and all antineutrinos are right-handed (spin +1/2); and (iii) the neutrino mass is zero [4]. The latter did not stand up to full scrutiny: with the discovery of neutrino oscillations in the 1990s, earlier suspicions and predictions that the neutrino indeed had mass, albeit extremely small, were experimentally underpinned [5].

However, while these results confirm that neutrinos have mass, their absolute mass scale was, and is, rather elusive: this is because neutrino oscillations are sensitive only to the difference in the squares of the masses (for a recent survey, see, e.g., Reference [6]). However, the knowledge of absolute values for the neutrino mass is of high importance for many aspects in particle physics, astrophysics, and cosmology (see, e.g., Reference [7]).

Of the various experimental approaches for determining the neutrino mass (for a recent survey see, e.g., Reference [8]), currently the most sensitive direct and model-independent method is the investigation of the electron energy spectrum of the tritium β-decay, H3→H3e+ + e−+ν¯e. The lowest upper limit for the electron neutrino mass from this type of experiments is that determined in the Mainz and Troitsk experiments. These were conducted independently and more or less in parallel during the 1990s up to the early 2000s, nearly contemporary to the aforementioned neutrino oscillation measurements. In both experiments, molecular tritium (T_2_) sources were utilized, namely condensed and gaseous tritium sources, respectively; and in both, the analysis of the tritium β-decay spectrum near the endpoint E_0_ = 18.6 keV was performed using a so-called MAC-E energy filter [9]. After some re-analyses, taking into account some initially omitted effects and all measurement data, the two groups reported “final” bounds for the electron neutrino mass (with 95% confidence limit) of *m*(${\bar{\nu }}_{e}$) < 2.3 eV/c^2^ [10] and *m*(${\bar{\nu }}_{e}$) < 2.05 eV/c^2^ [11], respectively. However, it was already clear then, from cosmological observations and calculations, that the mass limit was expected to be quite a bit lower. Thus, already during the latter stages of the Mainz and Troitsk experiments, a collaborative follow-up experiment—the Karlsruhe Tritium Neutrino experiment, KATRIN, to be hosted at the Tritium Laboratory Karlsruhe (TLK)—was proposed, and initiated, that should exhibit substantially improved measurement sensitivity.

## 2. Concepts of the KATRIN Experiment

The KATRIN experiment is designed to increase the sensitivity for measuring the neutrino mass by an order of magnitude to 0.2 eV/c^2^ [12,13]. The full setup of the KATRIN experiment, with overall length of 70 m, consists of a set of main component groups [13]; these comprise (i) the windowless gaseous tritium source—WGTS; (ii) calibration and monitoring systems to the rear of the WGTS—the “rear wall“ section; (iii) the transport section—for tritium flow reduction, ion removal, and adiabatic β-electron guidance; (iv) the spectrometer system—made up by a pre-spectrometer energy filter and the MAC-E type main spectrometer; and (v) the focal plane electron detector—a segmented-pixel device, optimized for best spatial resolution.

The core component of KATRIN—the β-electron source (WGTS)—is designed to run at a total activity of the source (i.e., the number of β-electrons per second leaving it) of up to 10^11^ Bq. Note that the active volume within the WGTS source is associated with the sensitive magnetic flux tube that connects the decay volume with the detector at the downstream end of the setup. It is defined by the confinement through the enclosing superconducting magnets [14]. The β-electrons are guided adiabatically (by means of magnetic fields) to the MAC-E type spectrometer for energy analysis, as in the previous experiments. For details of the overall instrument construction and design requirements, see Reference [13].

In order to manage (and minimize) the influence of statistics and systematics on the derivation of the neutrino mass, operating parameters need to be kept within very narrow margins [15]. In particular, the so-called column density *ρd* (see Section 4.3) is one of the experimental and theoretical key parameters in the evaluation of the KATRIN data. Besides this “global” parameter, the composition of the gas injected into the WGTS needs to be known with high accuracy. This latter aspect is one of the most important factors contributing to the KATRIN neutrino mass systematics. It is associated with the fact that the injected gas is not isotopically pure, meaning that it not only contains T_2_, but all other hydrogen isotopologues (DT, D_2_, HT, HD, and H_2_) are present as well, albeit mostly at relatively low concentrations. Particularly cumbersome for the determination of the neutrino mass is the presence of the tritiated isotopologues DT and HT, since these undergo radioactive β-decay as well:(1)$$T_{2}/\mathrm{DT}/\mathrm{HT}\;\to {\;}^{3}{\mathrm{HeT}}^{+}{/}^{3}{\mathrm{HeD}}^{+}{/}^{3}{\mathrm{HeH}}^{+}\;+\;e^{-}+\;{\bar{\nu }}_{e}.$$

The isotopic mass differences influence the kinetics of the decay (resulting in different recoil energy; and different initial and final state energy distributions of the parent and daughter molecules, respectively), and thus the energy partitioned to the β-electron [16]. Any small change of the tritium gas composition will manifest itself in non-negligible effects on the KATRIN neutrino mass evaluation [17]. This is because after the β-decay of any tritium isotopologue, according to Equation (1), the daughter molecule can end up in an electronic ground or excited state, each of them “broadened” by rotational and vibrational excitations; this gives rise to an isotopologue-dependent final state distribution. The excitation energy of the daughter molecule reduces the kinetic energy available for the β-electron, and thus the end point of the β-decay spectrum. As a consequence, one arrives at a composite superposition of spectra from all possible final states. Each individual spectrum has to be weighted by the probability to decay into a particular final state, with its spectral endpoint reduced by the corresponding final state energy. Therefore, it is of the utmost importance that gas purity and composition are monitored reliably, at all times.

Laser Raman spectroscopy—LARA—has been selected as the method of choice for this monitoring process, since it constitutes (i) a non-invasive and fast in-line measurement technique; and (ii) with suitable calibration, it provides quantitative concentration data, simultaneous for all gas constituents, specifically:(2)$$\mathrm{the}\;(\mathrm{relative})\;\mathrm{concentration}\;\mathrm{of}\;\mathrm{hydrogen}\;\mathrm{isotopologue}\;x,\;c_{x}=N_{x}/\sum_{i}N_{i}.$$

Here, the index *x* indicates a specific hydrogen isotopologue, with *N*_x_ being the number of molecules of any of the isotopologues within the WGTS (indices *i* and *x* ∈ T_2_, DT, D_2_, HT, HD, H_2_). Any sudden or severe changes in these concentration values can be signs of potential operational problems, in which case countermeasures can be implemented by KATRIN operation control.

Note that besides the concentrations, two derived parameters may be utilized during the neutrino mass evaluation, namely
(3)$$\mathrm{the}\;\mathrm{tritium}\;\mathrm{purity},\;\varepsilon_{T}=\left[N_{T_{2}}+\frac{1}{2}\left(N_{\mathrm{HT}}+N_{\mathrm{DT}}\right)\right]/\sum_{i}N_{i};\;\mathrm{and}$$
(4)$$\mathrm{the}\;\mathrm{HT}/\mathrm{DT}\;\mathrm{ratio},\;\kappa =N_{\mathrm{HT}}/N_{\mathrm{DT}}.$$

With the aid of *ε*_T_ and *κ*, the fractional activity for each of the tritium-carrying isotopologues can be parameterized (see, e.g., Reference [17]) for convenient incorporation into the neutrino mass analysis procedures. Note that the parameter *ε*_T_ is utilized when determining the total source activity. Experimentally, the count rate of β-electrons from the source, *S*_WGTS_ (as measured by activity detectors), is related to the column density and scales as (see, e.g., Reference [13]):(5)$$S_{\mathrm{WGTS}}=F_{S}\cdot N=F_{S}\cdot \varepsilon_{T}\cdot \rho d\cdot A.$$

Here, *F*_S_ is a proportionality function which includes—amongst others—the measurement properties of any specific electron detection method; N is the total number of molecules in the source volume; *ρd* is the column density; and *A* is the cross-sectional area of the source flux tube. The KATRIN experiment incorporates two activity monitoring systems (for details see [13] and references therein). One is situated in backward direction to the source, measuring current-induced Bremsstrahlung when the β-electrons impact on the rear wall—using β-induced X-ray spectroscopy (BIXS) [18]. The other is situated in forward direction and can monitor the β-electrons directly inside the flux tube—the forward beam monitor (FBM), based on a PIN-diode design [19].

In this context, it should be noted that the column density determines the scattering probability of electrons in the WGTS [20]. Unaccounted variations in the column density, *ρd*, as well as in the tritium concentration, *ε*_T_, and the relative concentration of the active gas isotopologues (T_2_, HT, and DT), can introduce distortions of the shape of the β-spectrum. As such, they contribute to the systematic error in the determination of the neutrino mass.

After the inception of the KATRIN experiment at the beginning of the 2000s, it took about a decade to construct, followed by a period of commissioning and testing. After long years of anticipation, KATRIN was inaugurated in June 2018, and the experiment is fully operational now. Initially, it was run at greatly reduced tritium content in the WGTS [21], thus keeping radioactive loads at a minimum, to allow for potential repairs and system modifications. Full KATRIN data taking commenced, after tritium ramp-up, during the first quarter of 2019. Results from this early neutrino mass measurement campaign have now been published, yielding an upper bound for the electron neutrino mass of *m*(${\bar{\nu }}_{e}$) < 1.1 eV/c^2^ (90% confidence limit) [22].

Here, we report on the performance of the LARA gas monitoring system, comparing pre-KATRIN results and those from actual KATRIN data-recording campaigns (which needed precise real-time analysis), and ascertaining whether all KATRIN requirements are effectively met. In addition, we discuss various in-line chemical reaction processes which alter the circulating gas composition, and which might therefore adversely impact the neutrino mass result.

## 3. Setup and Methods

It is beyond the scope of this publication to describe all system components in detail; here, emphasis will be on WGTS operation and, specifically, on the monitoring of its tritium gas composition.

### 3.1. Gas Circulation within the WGTS Loop

The WGTS—the tritium β-electron source—may be seen as the core component of the KATRIN experiment. Tritium gas (predominantly in the form of molecular T_2_) is continuously injected at the center of the 10 m WGTS beam tube and pumped out through its ends, maintaining a constant pressure profile between the inlet and the outlet(s). Because of the gas flow nature of the tritium source, and for tritium recuperation, the WGTS is incorporated into a loop system. This loop system is subdivided into two conceptual parts, the “inner” and “outer” loops. Here, only the inner loop part is described in more detail, to facilitate the discussion of various gas composition and circulation issues, which impact on the reliability of the KATRIN experiment.

Note that the components of the outer loop are not directly related to KATRIN, but, by and large, form part of the TLK infrastructure, including, as the most important unit, exhaust gas cleanup; isotope separation; and isotope storage and transfer, but also incorporating the capabilities for tritium analytics and accountancy [23].

The concept of the inner loop, together with the WGTS itself, is shown in Figure 1a (see also Reference [24]). From a reservoir (buffer vessel at a pressure of the order 200 mbar) the gas flows through the laser Raman (LARA) cell to a pressure- and temperature-controlled buffer vessel (pressure <20 mbar), from which it passes through a transfer tube and capillary to the WGTS injection chamber. Note that the WGTS beam tube is kept at cryogenic temperature of about 30 K, using an ultra-high stability two-phase neon cooling system [25]. For special measurements, it can also be operated at higher temperatures, using nitrogen (80 K) or argon (100 K), respectively. The WGTS tube is also imbedded into a superconducting magnet system, generating fields of up to 3.6 Tesla on the beam tube axis. The gas is extracted at both ends of the beam tube via differential pumping ports (for clarity, not all pumps and vacuum lines for all ports are drawn in the Figure), and is transported back to the reservoir buffer vessel (using a compressing circulation pump arrangement). On its way, the gas traverses a palladium membrane permeator [26]. The role of this permeator is to remove all impurities from the circulating gas to allow only the hydrogen isotopologues to pass through.

About 1% of the gas flow is bled off into the outer loop system for cleanup and tritium recovery; of course, to maintain a constant pressure and flow structure in the loop, cleaned “pure” T_2_ (>95%) is added at the same rate from the outer loop structure into the reservoir buffer vessel.

When inspecting the inner loop—including the WGTS—more closely, it becomes clear that the analysis of the gas composition and flux are rather challenging. In Figure 1b, the loop segment from the LARA cell to the WGTS injection chamber is shown. Evidently, one encounters a huge variation in pressures, flux conditions (associated with pressure, diameter, and length of tubing) and temperatures, with the typical numerical values included in the Figure.

The main difficulty in determining the KATRIN-required parameters *c*_x_, and the derived *ε*_T_ and *κ* (see Equations (2)–(4)) is that the associated LARA measurement is localized far away from the tritium column within the WGTS beam tube. Furthermore, the pressure, flow, and temperature conditions are very different at the measurement site and inside the WGTS, ranging from ~200 mbar, viscous flow and ~300 K; to ~10^−3^ mbar (or less), molecular flow and ~30 K, respectively. However, the Raman measurements cannot easily be carried out at any desirable location within the complete loop system, specifically inside the WGTS beam tube where one would like to determine the *c*_x_, *ε*_T_, and *κ*. First, the low pressures within the WGTS (and, therefore, low molecular number densities) make it next to impossible to directly measure the gas composition (with relative partial pressures of the constituents varying over two to three orders of magnitude). Second, Raman measurement components—e.g., a cell—would block the line of sight of the beam tube and thus impede the proper functioning of the WGTS. Although, with the advancement of Raman measurement equipment, it is now within the realm of feasibility to overcome the former problem, the second impediment would still persist, and therefore render continuous measurement during KATRIN operation impossible.

However, it has been demonstrated during the test phases for the Raman equipment and the gas circulation loops, as well as during the first KATRIN data taking campaign, that the derived parameter values—associated with the LARA measurement data, recorded at the current location in the loop—seem to be adequate for the determination of the neutrino mass, in reference to recent modeling of the gas flow conditions in the WGTS [27].

### 3.2. The Laser Raman System

The concept of the laser Raman (LARA) monitoring system is shown in Figure 2. The setup follows closely that originally proposed in the KATRIN Design Report [13]; its capabilities were successfully demonstrated in 2008 during the “TILO” (Test of Inner LOop) experiment [28], to ascertain that the inner loop functioned correctly and fulfilled all design specifications. It should be noted that those tests were performed without the WGTS, which was still under construction at that time. Since then, a number of technical and procedural improvements have been incorporated which have augmented the sensitivity and precision of the LARA measurements.

While already the first LARA setup, dating from 2005/6, provided satisfactory results, improvements since then afforded that the KATRIN requirements could be met and, in some cases, exceeded. These improvements include (i) the upgrade with improved components—specifically the excitation laser and the CCD array light detector; (ii) the implementation of a double-pass configuration—nearly doubling the available Raman excitation laser intensity; (iii) fully quantitative Raman light intensity calibration—see, e.g., Reference [29] and Section 3.3; and (iv) the development of a bespoke data acquisition and evaluation software package—for its concepts, see Reference [30] and Section 3.4). The equipment components of the actual KATRIN LARA setup used in this study are summarized in Table 1.

The Raman gas cell itself is a tritium-compatible component, which was designed and used in the very first attempts of Raman spectroscopy of hydrogen at TLK [31]. In particular, the cell needs to be compliant with tritium radioactivity requirements (e.g., gas leakage), and the optical windows and their coatings need to be as resistant as possible against β-radiation damage.

The cell is mounted in-line in the inner loop for tritium circulation, between the two main buffer vessels of the loop (see Figure 1). As a consequence, the cell is situated within the secondary (glove box) enclosure; access for laser radiation and Raman light is through anti-reflection coated windows in a bespoke “appendix” extension of the glove box. Note that the Raman cell can be temporarily exchanged with a calibration assembly (see the top-right in Figure 2), which incorporates a Raman intensity standard (NIST, *SRM 2242*), for in situ absolute light intensity calibration; for a more detailed description, see Section 3.3 below.

A continuous wave (CW) green (λ_L_ = 532 nm) DPSS Nd:YVO_4_ laser (Laser Quantum, *Finesse*), operated at an output power level of *P*_L_ = 2–3 W, is utilized in the current setup. The laser radiation is guided, filtered, and focused into the Raman gas cell by a long focal-length lens, generating an elongated, near-cylindrical interaction volume of about 6 mm in length and a beam waist of ~100 μm. In variation to the original design, now the laser beam is back-reflected through the cell, using a combination of a lens (with the same focal lens as utilized for the input coupling) and a dielectric mirror. This replicates the primary excitation volume, approximately doubling the Raman response probability. Note that the back-reflected laser light is directed out of the beam axis by the optical Faraday isolator; this reflection is beneficially used to monitor the laser power and potential laser beam walk during long-term operation.

The Raman light from the elongated excitation volume is collected at a right angle to the laser excitation axis (i.e., 90° scattering geometry), and is focused onto a “slit-to-slit” shaped optical fiber bundle (for its dimension, see Table 1). Note that a set of two plano-convex lenses is used to provide 1:1 imaging of the Raman interaction volume onto the entrance of the fiber bundle, with an acceptance angle matched to the numerical aperture of the fiber. Prior to entering the spectrometer (PI Acton, *HTS*, with *f*# = 1.8) for spectral analysis, the collected light is cleaned for any residual laser radiation by a long-pass filter (Semrock, *RazorEdge 532nm*). The spectrometer’s range and resolution are set so that all six hydrogen molecule isotopologues are resolved and recorded simultaneously by the high-sensitivity CCD array detector with 512 × 2018 pixels (Princeton Instruments, *Pixis 2k*).

As indicated in the lower right of Figure 2 system control, recording of operating parameters, Raman spectrum acquisition, and real-time data treatment and analysis are under the control of an integrated, dedicated *LabVIEW* program (for a brief outline, see Section 3.4). The numerical values for *c*_i_, *ε*_T_, and *κ* are extracted from the spectra and transmitted to the KATRIN database, synchronous with the scan protocol.

### 3.3. Aspects of Calibration

As just stated, the numerical values for *c_x_*, *ε*_T_ and *κ* are all derived from the analysis of the Raman signals. As in all spectroscopic methods, the information about the analyte is provided in the form of the observed (measured) spectral light intensity, as a function of wavelength (or wavenumber). The measured Raman response of a probed molecule to the laser light stimulus is given by the (simplified) expression:(6)$$R_{\Delta \upsilon }\left(J\prime \prime \right)=k_{\lambda }\cdot \lambda_{L}^{-1}\cdot \lambda_{s}^{-3}\cdot \eta \left(\lambda_{s}\right)\cdot \Phi \left(\Delta \upsilon ;J\prime \prime \right)\cdot N_{x}\left(T\right)\cdot I_{L},$$
with:$k_{\lambda }$ = unit normalization constant$\lambda_{L}$ = laser wavelength$\lambda_{s}$ = Raman line wavelength$\eta \left(\lambda_{s}\right)$ = spectral sensitivity of the LARA system at the Raman line position$\Phi \left(\Delta \upsilon ;J”\right)$ = transition probability function for Raman line, for initial rotational level *J*″$N_{x}\left(T\right)$ = number of molecules, in their initial energy state, at gas temperature *T*$I_{L}$ = incident laser power density

Note that for the KATRIN analysis, at present, only lines for the vibrational transition Δυ=+1 are included, i.e., values for the Raman response $R_{1}\left(J\prime \prime \right)$ for the respective O_1_(*J*″)-, Q_1_(*J*″)-, and S_1_(*J*″)-branch lines are required. In order to quantify the contributions from the individual isotopologue mixture constituents, (i) exact absolute calibration in wavelength and light intensity response is required, and (ii) a quantitative link between the observed Raman signals and the (unknown) molecular number densities needs to be established.

#### 3.3.1. Wavelength Calibration for *λ*_s_

Since the spectrometer configuration for the KATRIN LARA system is chosen to cover the complete range of spectral contribution from all six hydrogen isotopologues (~570–710 nm), the spectrometer wavelength setting can be “fixed”, i.e., no grating motion is required any longer, in principle, after the initial setup. Then, well-known emission lines from hollow-cathode calibration lamps are recorded, from which the association of each pixel with wavelength can be determined once and be fixed for all future measurements. However, it seems always prudent to carry out occasional validity rechecks.

#### 3.3.2. Spectral Intensity Response Function, *η*(*λ*_s_)

On its way from the Raman excitation volume to the CCD array detector, the light passes through a substantial number of optical components, including various lenses, the Raman edge filter, the optical fiber bundle, and the spectrometer grating, to name but the most prominent elements. All exhibit their individual, wavelength-dependent transmission/response functions. In addition, the CCD detector itself possesses wavelength-dependent photon sensitivity. Overall, this means that the relation between the signal from the excitation volume and the actually recorded signal incorporates a complex convolution of response functions, complicated even further by the fact that the Raman signals themselves and light propagation through certain optical components are polarization-dependent as well. Unfortunately, in most cases, only generic response functions are provided by instrument manufacturers; individual items may substantially deviate from those functions.

The most sensible solution for this conundrum is to place a calibrated emission standard at the position of the Raman excitation location and experimentally measure the overall response; this is the only way to determine an absolute response function *η*(*λ*_s_) of percent-level correctness. In the case of the KATRIN LARA system, such a task proved to be extremely difficult, in particular because of the restricted access to the Raman cell location within the secondary-enclosure glove box. The method selected in the end is based on a fluorescence calibration standard (NIST, *SRM2242*), which is incorporated into a “mock-up” assembly for direct replacement of the actual Raman cell (see the top-right in Figure 2). In brief, the SRM standard is lowered into the laser beam path so that fluorescence is generated at exactly the same position as the ordinary Raman excitation in the gas. The fluorescence from this same-size volume passes through the Raman light collection system. The recorded fluorescence spectrum is then compared to the NIST SRM2242 reference data, thus providing an absolute intensity calibration including the full optical detection system. Details of the calibration setup and procedure can be found in references [29,32].

#### 3.3.3. Linking the Measured Raman signal to the Particle Density

With the majority of parameters in Equation (6) known—either measured directly (i.e., *λ*_s_, *λ*_L_, and *I*_L_) or derived in a calibration measurement (i.e., *η*(*λ*_s_))—it can be reduced to the form
(7)$$R_{\Delta \upsilon =+1}\left(J\prime \prime \right)=F_{\mathrm{known}}\cdot \Phi \left(\Delta \upsilon =+1;J\prime \prime \right)\cdot N_{i}\left(T\right),$$
in which all known quantities are subsumed into the global correspondence factor *F*_known_. This leaves the Raman transition probability function and the particle density as the “unknowns”. There are two conceptual approaches of how to gain access to the desired isotopologue concentrations using Equation (7); ideally, one should apply both, if possible, which would provide a consistency test. However, neither of the two methodologies is easy to implement a priori.

In the first approach, one derives the transition probabilities for the respective ro-vibrational Raman lines from theoretical (ab initio) calculations. However, the relevant transition probabilities may not have been published, nor is it in general straightforward for an experimentalist to carry out the required calculations, even if some basic code were available. For the hydrogen isotopologues discussed in this paper, the values for all individual Raman lines have been kindly calculated by R.J. LeRoy [33].

In principle, one also might be able to gain access to the Raman transition probabilities experimentally, making use of the so-called method of Raman depolarization (see, e.g., reference [34]). We have done this for a substantial number of the required Raman lines [35], and the values are in very good agreement with LeRoy’s theoretical values (in most cases, much better than 5%). For the results presented here, this methodology has been applied; example data are given for relevant, observed Raman transition lines in Section 4.3 below.

It also should be noted that for the determination of concentrations from spectral features (here, the Q_1_-branches of the relevant isotopologues), care has to be taken to appropriately treat potential line overlaps originating from different molecular species. This aspect is discussed in detail in Section 4.2.

The second approach is akin to the well-known analytical method of dilution chemometrics; from exactly known relative concentration mixtures, calibration curves are constructed, associating a known reference concentration with the recorded signal. This has been done for the non-radioactive isotopologues of hydrogen—H_2_, HD, and D_2_—in a measurement setup at TLK, dubbed “HyDe” [36,37]. In brief, for the calibration of LARA data against known concentration data, two combined datasets are necessary, namely (i) a binary dataset with only the initial isotopologues not in thermal equilibrium; and (ii) a tertiary dataset with all three corresponding isotopologues in thermal equilibrium. The agreement between the two methods was excellent, mostly <3% for the Q_1_-branch signals.

The approach is much more complex when adding the isotopologues with radioactive decay, i.e., T_2_, DT, and HT, since the mixing ratios may not remain constant over the measurement period. On one hand, this is caused by the β-decay of tritium (resulting in a loss of about 1% over the period of one month). On the other hand, tritium-mediated reactions at the steel walls of the loop (see Section 4.5 for relevant details) may introduce measurable, but uncontrolled amounts of reaction products, which therefore affect the initial concentration of the parent molecule(s) on the time scales required to generate full calibration curves. A tritium-compatible mixing system has now been constructed—conveniently named “TriHyDe”—with which all six hydrogen isotopologues can be tackled [38]. In spite of the aforementioned complications, the first results look rather promising. Foremost, the HyDe calibration for H_2_:D_2_ mixtures could be replicated, yielding agreement to within 2–3%. Measurements of binary and tertiary mixtures containing tritium are now almost complete, and evaluation of the data is ongoing. Just as an example, for the binary mixture D_2_:T_2_, the calibration factors are accurate to less than 2%. The description of the related TriHyDe system setup and results from the full calibration sets of all six isotopologues will be subject to a forthcoming publication.

### 3.4. Automated Data Processing

For the recording of the spectral data and the evaluation of the raw data towards the desired parameters, to be transferred in near real-time to the KATRIN control, a bespoke suite of modular software routines is used (*LARAsoft*, written in *LabVIEW 2014* for flexibility and conformity with industry-standard control of the LARA instrumentation). The current second-generation software package has evolved from the earlier, integrated procedure described in reference [30]. The software suite now comprises four hierarchical control and evaluation modules; the sequence and purpose of the various program steps/modules is shown in the conceptual diagram Figure 3.

#### 3.4.1. LARA System Control

As stated earlier, the operation and monitoring of the majority of the LARA equipment is under the control of *LARAsoft*, including (i) setting up and monitoring the Raman excitation laser, as is monitoring the laser power passing through the Raman cell, and potential beam walk; and (ii) setting up and pretreating the spectral data accumulated by the 2D CCD detector of the spectrometer (some issues associated with the timing of spectra recording and evaluation of Raman data, with reference to KATRIN run-time tags, are discussed in Section 4.4).

#### 3.4.2. Detector Readout, Sensitivity Calibration and Spectrum Generation

The “raw” light signal data accumulated on the 2D CCD chip undergo partial on-chip pixel binning prior to readout (in the non-dispersive direction of the detector chip), in order to accelerate readout and to reduce the contribution from readout noise. For the measurements reported here 20 bins of 20 pixels each were utilized. These binned spectral data sets are then treated for a variety of effects, to improve the quality of the final Raman spectrum. These include, as described in reference [30], (i) the removal of cosmic rays (to avoid potential false feature identification; note that for this procedure at least two subsequent Raman data sets are required); (ii) spectral intensity correction (utilizing a 2D calibration map generated from supplemental SRM2242 measurements); (iii) spectrometer astigmatism correction (shifting the aforementioned binned chip segments to line up in the non-dispersive direction); (iv) summing up all bin segments (to generate the total-signal Raman spectrum); and (v) the removal of any background (mostly fluorescence induced by the excitation laser in optical elements)—note that the removal procedure is based on the rolling-circle filter methodology (see, e.g., [39,40]).

#### 3.4.3. Spectrum Evaluation

For the quantitative evaluation of any spectrum, its dataset is first linked to an accurate wavelength scale, using a generic pixel vs. wavelength reference function; this has been generated offline from recorded spectra of atomic emission line sources. However, normally the wavelength calibration is only utilized for feature identification and visualization; all numerical procedures described here are carried out on pixels. The KATRIN-required parameter values are extracted from the spectra utilizing the Q_1_-branch signals of the six hydrogen isotopologues. The preparation and evaluation of these six components is carried out in two steps.

First, any interfering overlap contributions need to be eliminated. This is because the S_1_(*J*″)- and/or O_1_(*J*″)-lines—whose intensities in general are a factor 20–100 smaller than the Q_1_(*J*″)-lines—of a highly abundant isotopologue may still be intense enough to contribute measurably to the Q_1_-branch of a minor isotopologue, if the features coincide spectrally. The underlying procedure is outlined in more detail in Section 4.2.

Second, the total intensity of the various Q_1_-branches (whose rotational lines are unresolved when using the KATRIN LARA instrumentation) is calculated. For this, appropriate spectral intervals are set so that the entire range of contributing Q_1_(*J*″)-lines is covered; then the intensity contributions are integrated over the respective, individual intervals and stored for further evaluation (using a bespoke routine *ShapeFit* [30]).

#### 3.4.4. Calculation of Parameter Values for the KATRIN Experiment

From said integral Q_1_-branch intensities the KATRIN parameters are calculated, combining Equations (2)–(4), and (7). The input to the latter is the Raman transition probability Φ from [33]; the probabilities are stored in a call up table. In addition, for all parameter values error propagation is carried out as well, following the procedures outlined in reference [32]; important aspects of the parameter uncertainties and results relevant to KATRIN are summarized in Section 4.4. Finally, all calculated parameter values and associated uncertainties are transferred to the KATRIN database; these are linked to the time-tags of the advancement step within a particular KATRIN data taking run, to be used in the neutrino mass analysis procedure. Note also that in addition to the ADEI data, all raw (but binned) spectral data from the detector chip readout are stored offline, enabling full re-analysis if necessary, in the case that some aspects of the KATRIN analysis may be revised over the lifetime of the experiment.

## 4. Results

As stated earlier, the KATRIN experiment has been fully operational since early 2019, and a wealth of data has been collected, both for system performance measurements as well as for first, actual neutrino mass data recording. The aforementioned tritium-related parameter values rely on Raman spectroscopy measurements, in which all hydrogen isotopologues circulating in the loop are monitored.

Up until the commencement of the first neutrino mass campaign in the first half of 2019 (KATRIN “KNM1”), the gas composition inside the WGTS was kept “tritium-lean” (*ε*_T_ ≈ 1 %), meaning that it was extremely different from the conditions for standard KATRIN operation [13]. Nevertheless, those “first tritium” (FT) measurements provided valuable information as well, since important aspects of changes in gas composition could be studied, while keeping radioactivity at a minimum.

Thus, in the following discussion of results, both scenarios—low-tritium and nearly-pure-tritium content—are contrasted. In addition, the KATRIN system results are compared to earlier “LOOPINO” measurements [41], in which the circulating gas loop was operated without the WGTS and without the permeator unit (although operating conditions were kept as close as possible to the later KATRIN measurements). The specific operating conditions for the three cases are summarized in Table 2.

### 4.1. Spectra for the Different T_2_ Circulation Scenarios

As was just pointed out, three overall different operating conditions for tritium gas compositions were measured and studied in detail, namely KATRIN “KNM1” and “LOOPINO” for high-tritium content (>95%) and “FT” for low-tritium content (<1%). A set of typical Raman spectra is shown in Figure 4. Note that all measurements described here were carried out for a total gas pressure within the LARA cell of *p*_RC_ ≈ 150–200 mbar; this is the customary pressure at which the supply buffer vessel is kept under normal KATRIN operating conditions.

#### 4.1.1. LOOPINO Spectra

With reference to Figure 1a and the setup sketch in [29], the loop tubing from the pressure-controlled buffer vessel was directly coupled to the circulation pump, and the permeator was bypassed. Note that the WGTS was not yet on site, and not all of the outer loop infrastructure was completed at the time of the LOOPINO measurements; also, in contrast to the actual KATRIN setup, the complete loop was at room temperature. This simplified configuration was beneficial for the understanding of some of the chemical processes occurring within the inner loop system.

For the long-term functionality tests, the LOOPINO system was filled once with an initially “pure” high-T_2_ gas mixture (for the composition of the reservoir gas prior to injection see Table 3). In Figure 4a, two “snapshot” spectra are shown, recorded at the beginning of the run (during day 1) and towards the end of the run (after 55 days), respectively; the spectral traces represent averages over 20 individual CCD data accumulations of 60s duration each. The following observations can be made.

First—as expected—the spectrum is dominated by features of T_2_, with the strongest being the (unresolved) ro-vibrational Raman band Q_1_(*J*″), but with (resolved) O_1_(*J*″)- and S_1_(*J*″)-lines visible as well.

Second—as expected—substantial amounts of HT are generated (increasing from ~4.5% to ~6.5%), associated with the aforementioned surface-mediated reactions of tritium with H_2_O on and hydrogen in the steel walls.

Third, over the 55-day period of the measurement, the amount of DT does not noticeably change (within the measurement accuracy) because of the lack of an additional D/D_2_ source.

Fourth, over the measurement period, a significant accumulation of tritium-substituted methane compounds is observed; the most pronounced vibrational Raman bands are annotated in the Figure. These molecules are also due to surface-mediated reactions, namely of tritium with carbon atoms liberated from the steel walls (predominantly by β-electron bombardment); see Section 4.5 for details.

Normally, these “contaminants” would not be visible above the detection threshold of our LARA system, because they would be removed from circulation by the permeator. But due to its absence in the LOOPINO loop, any generated chemical compound can accumulate. The two surface-mediated reactions, i.e., the extraction of hydrogen and carbon, will be discussed in Section 4.5.

#### 4.1.2. First Tritium (FT) Spectra

For the very first full operation of the KATRIN experiment with tritium, the pre-determined gas mixture admitted into the inner loop (including WGTS and permeator) only contained a small amount of tritium (for the reservoir gas composition prior to injection see Table 3). Thus, the observed Raman spectra are significantly different from the high-T_2_ spectra.

A typical “snapshot” spectrum for this operating scenario, recorded after two days of operation, is shown in Figure 4b; note that the same data recording times and spectra-averaging procedures were used as during the LOOPINO runs. The following features can be identified in the spectrum.

First, in line with the gas composition, the features of D_2_ dominate, with the expected intensity ratios between the Q_1_(*J*″) branch and the O_1_(*J*″) and S_1_(*J*″) lines.

Second, because of the action of the permeator, the ratios between D_2_, DT and T_2_ approach equilibrium very quickly, with the latter nearly absent, due to the dynamic chemical equilibration.

Third, as in the previous case, hydrogen is liberated from the system’s steel walls, and as a consequence, a steadily increasing amount of HD is observed over the measurement period. Because of the very much lower content of tritium, the also expected HT is not detectable above the noise level.

Fourth, no isotope-substituted methanes are observed; in contrast to the LOOPINO test experiment, here, the permeator removes them efficiently from the loop, so that they cannot accumulate to measurable amounts.

Finally, it should be noted that the weak spectral feature observed just below 580 nm can be attributed to Raman scattering in the fused silica window(s) of the cell. However, it also should be noted that contributions from this (unwanted) Raman and fluorescence background from the cell windows strongly depend on the alignment of the cell within the laser beam path. The magnitude of this background changes substantially with only minutely different orientation of the cell, or when changing the cell in between measurement runs, and thus may be observable, or not.

#### 4.1.3. KATRIN (KNM1) Spectra

During the first full KATRIN neutrino mass run (denoted KNM1) during the first half of 2019, the gas mixture admitted into the system was, as prescribed, of high-T_2_ content (for the reservoir gas composition prior to injection, see Table 3). Once more, the spectra shown here are averages over 20 individual accumulations; see Figure 4c.

As expected from a tritium-rich mixture (which, in fact, is very similar in its initial composition to the LOOPINO gas supply), the spectra from the LOOPINO and KNM1 measurements closely resemble each other. All hydrogen isotopologues are observed; based on the initial concentrations in the feed and the estimated liberation of hydrogen from the steel walls, they exhibit roughly the expected relative Raman intensities.

The main difference is the lack of spectral features from methane; this is hardly surprising since—as in the FT and KNM1 campaign scenario—the permeator prevents them from accumulating in the loop over time, but rather extracts them from the circulating gas, and, in addition, methane condenses in the injection capillary (held at ~30 K). It should be noted that, when comparing the temporal evolution of the individual isotopologue compositions for the LOOPINO and KATRIN scenarios, different rates are observed for the two. Again, this is not unexpected, since LOOPINO comprised a “closed” loop without gas exchange, while for KNM1, the bleeding-out from the permeator and feeding-in into the reservoir vessel (balanced for constant pressure and flow) constitutes an “open” loop. The evolution of concentrations over time is discussed in more detail in Section 4.3.

### 4.2. Disentangling Spectral Overlap Features

Before the isotopologue concentrations can be extracted (based on the quantitative analysis of the integral Q_1_-branch intensities), one particular issue needs to be addressed: overlapping features in the spectra can interfere with the determination of the true concentration values.

When inspecting the spectral traces in Figure 4, specifically those from the LOOPINO and KNM1 measurements, the wavelength region near ~623 nm (equivalent to a Raman shift of ~2730 cm^−1^) commands particular attention. Here, the S_1_(*J*″ = 2) line of T_2_ coincides with the Q_1_-branch region of DT; given the resolution of our spectrometer, these cannot be separated.

Because of its very high T_2_ concentration, its Raman lines dominate in the spectra; even its S_1_- and O_1_-branch lines with low transition probability (a factor 50–100 less than the Q_1_-branch lines) have sufficiently high intensity to be clearly visible in the spectra. This means that they are, or can be, of similar intensity to that of the Q_1_-branch of DT, with a relative concentration only in the percent regime in comparison to T_2_. Hence, in order to quantitatively extract the concentration for DT, said spectral overlap needs to be disentangled into the contributions from T_2_ and DT.

It should be noted that, while there are quite a few similar overlap regions in the spectra containing all hydrogen isotopologues, for the three relative concentration scenarios covered here (with very high and very low T_2_ content), only the aforementioned overlap plays a significant role for the KATRIN experiment and therefore needs to be dealt with.

The procedure applied here is straightforward in principle but requires a number of analytical steps. In addition, several theoretical and experimental parameter values need to be known in order to apply the line intensity Equation (6) for quantitative analysis. The procedural steps of how to separate the interfering Raman signals are shown in Figure 5.

In the first step, (ideally all) resolved and isolated S_1_(*J*″) and O_1_(*J*″) lines of T_2_ in the recorded LARA spectrum (top-left data panel) are fitted using the appropriate *ShapeFit* line profiles. Note that for these near-symmetric S_1_(*J*″), O_1_(*J*″) lines, the spectrometer resolution profile—based on the slit width and the intensity profile from the fiber bundle—may alternatively be used (see the top-right panel in Figure 5). The so determined line intensities are then compared with the theoretical Raman transition line strengths (center-right data panel in the Figure), in conjunction with the thermal level population at T = 298 ± 2 K, which best fits the experimental line intensities. Note that this temperature is consistent with the gas temperature T_BV_ = 299–301 K, measured at the loop buffer vessel about 80 cm upstream from the LARA cell.

Based on these fit results, the unknown experimental intensity of the overlapping S_1_(*J*″=2) line can be calculated. Thus, a complete Raman spectrum for the Δ*J* = ±2 branches can be modeled (center-left data panel in the Figure), which then is subtracted from the original LARA spectrum. The result of this operation is that only the Q_1_-branches for T_2_ and DT remain (bottom-left data panel in the Figure). Finally, the integral branch intensity is determined from which actual concentrations can be calculated exploiting Equation (6). Related numerical data are collated in Table 4.

It should be noted that the principle of the disentanglement example described here is applicable to all other spectral overlaps potentially encountered in Raman spectra of different relative hydrogen isotopologue composition. An example would be “first tritium” gas mixture, with its dominant D_2_ concentration, whose spectrum is included in Figure 4. There the O_1_(*J*” = 3) line of D_2_ interferes with the Q_1_-branch of DT, and the S_1_(*J*” = 4) line of D_2_ interferes with the Q_1_-branch of HT. However, in this specific case, the full procedure described above is not required, since the features are partially resolved and simple *ShapeFit* application did suffice to remove the D_2_ interference.

A full list of potential interferences for all hydrogen isotopologues, and parameter values necessary for their efficient removal, can be obtained from the authors.

### 4.3. Temporal Evolution of the Concentrations of T_2_/DT/HT

As has been pointed out a few times already, one expects that the gas composition circulating in the loop may change over time. Of course, the “time constants” for each of the three scenarios (LOOPINO, FT, and KNM1) will be different because the operating conditions are very different for each of them. In the first case, a closed loop without permeator allows for the accumulation of any “contaminant” generated during the circulation. In the latter two cases, the loop is open, i.e., a small fraction of the circulating gas is extracted from the permeator unit for cleanup, while the same amount of cleaned gas is reinjected into the buffer vessel of the loop (see Figure 1a). However, in all three scenarios, one encounters “sinks” and “sources” for any of the isotopologues, as well as additional chemical compounds which might be generated during circulation (like the aforementioned methanes).

While, in principle, it would be feasible to calculate said time constants in the concentration evolution of the different gas components, in practice, this is rather complex, in particular for the full KATRIN loop, including WGTS and permeator. For example, the removal and injection rates from the permeator and into the buffer vessel are preset, as well as the flow rate. Any extraction of hydrogen and carbon from the walls and deposit of tritium—although known in principle—would require full modeling of the loop, including the varying pressure, flow, and temperature conditions. This has not been done for the data presented here, but only conceptual arguments have been utilized.

In this section, we only address the variation of the gas composition for the ramp-up phase for KNM1 (about two weeks in duration). On one hand, it is the neutrino mass measurement mode which is of highest interest and importance (i.e., long, stable operating conditions); however, on the other hand, during the ramp-up phase, drastic changes in the flow rate, and thus column density in the WGTS, allow one to appreciate the temporal reaction of the loop system to “sudden” variations in operating conditions and how long it may take until the system has “quieted down” again. In addition, one can evaluate whether the precise knowledge about the gas composition and its variation are still meeting the KATRIN requirements for utilizing the associated neutrino mass measurement data. This is of utmost importance since the abundance of the individual tritium isotopologues T_2_, DT, and HT, as well as the associated parameters ε_T_ and κ, are required for accurate and reliable evaluation of the measurement data (a detailed discussion of the influence of individual isotopologues and their state distribution onto the neutrino mass can be found, e.g., in reference [17]).

For an appreciation of the relative abundance data for the individual tritium isotopologues, shown in Figure 6, it is worthwhile to briefly contemplate the flow rates encountered during KNM1 and any future KATRIN measurements. With reference to Figure 1b, the actual flow of gas into the WGTS is determined at the pressure-controlled buffer vessel from where the gas mixture travels through the tubing system, for injection at the center of the WGTS. This flow rate changes when the column density, *ρd*, inside the WGTS is chosen at any desired value in the full range 0% < *ρd* < 100%. For example, the flow rate at full nominal column density (*ρd* ≡ 100%) was measured at 89.2 sccm; for a column density of only *ρd* = 25%, this value reduced to 13.3 sccm (note that for KNM1, for long periods, the system was run at *ρd* = 25%, for operational reasons).

Taking into account the buffer vessel volumes and the flow rate values, one full circulation of the KATRIN gas volume through the loop system exhibits characteristic times of ~8 min (for *ρd* = 100%) and ~37 min (for *ρd* = 25%), and even longer for the lower column densities utilized during the initial ramp-up phase. If one also considers the fact that during each circulation, 1% of the gas volume is exchanged to/from the outer loop reservoir, one expects time constants of the order of hours until the loop has reached a new equilibrium again, following a sudden change. This behavior is clearly visible in the plots for the temporal evolution of the molecular concentrations.

The system response to the (sudden) ramp-up changes, of the order 30–60 min, clearly reflects the contribution to the apparent concentrations from “sinks” and “sources”. For example, at the first step from the gas being static to the flow for a relative column density of *ρd* = 5%, the concentration first decreases rapidly, approximately replicating an exp(−x/τ_1_) behavior, and then undergoes “recovery”, exhibiting a functional dependence of the approximate form (1 − exp(−x/τ_2_)), with a different time constant. The inverse, mirror-like behavior is observed for the concentration of HT.

To a certain extent, this confirms the interdependence of the two isotopologues T_2_ and HT in the loop circulation, where the latter is generated by the extraction of hydrogen from the steel tube walls and “catalytic” exchange reactions with T_2_ molecules, whose relative concentration diminishes as a consequence. While during LOOPINO this was a smooth, one-way progression, for the “open” KATRIN loop, initial response is counteracted by the extraction of fractions of the gas mixture and insertion of “pure” T_2_ into the loop. When waiting long enough, a T_2_/HT equilibrium should evolve; this would be characteristic for a given flow rate (or column density). In the “quick” ramp-up scenario implemented during KNM1, the system settings already change again before the particular equilibrium is reached.

The behavior just described repeats itself in roughly the same manner at each of the steps when setting a new relative *ρd* value. It is worth mentioning that similar trends are also observed for DT, albeit on a much reduced scale of relative variation (see the central panel in Figure 6). This should not really be surprising, since there is no major “source” for deuterium, in contrast to hydrogen which is extracted from the steel walls.

It also should be noted that the description of the temporal behavior in the evolution of the molecular concentrations is most likely over-simplistic: a single-source single-sink scenario is assumed, which in all likelihood is not sufficient, and the interplay of various generation/removal channels in the loop is much more complex.

Furthermore, we would like to point out that regular changes in the concentration values measured by KATRIN LARA are expected (and observed), even for operating at constant column density over a complete KATRIN run of several weeks. This is associated with the fact that the reservoir gas is replenished regularly, on average once per week.

While every attempt is made to keep its composition as constant as possible, after cleaning of the gas extracted from the loop, the relative content of all six isotopologues varies ever so slightly. Of course, the loop gas composition reacts to this when the supply gas mixture is different, tending to new concentration equilibria. The data from the KATRIN Raman monitoring provide accurate information about these slow changes over time.

Finally, we would like to note that temporal evolution data for the molecular components in the LOOPINO campaigns may be found elsewhere [41,42]. Some selected measurement data for the FT campaign are presented in [21].

### 4.4. Precision and Stability

In any long-term experiment, careful monitoring is normally required which ascertains that a certain measurement parameter stays within predefined limits; this applies to KATRIN as well.

For KATRIN, one of the key parameters is the tritium purity, *ε*_T_, which needs to be monitored with a statistical uncertainty ${\left|\Delta \varepsilon_{T}/\varepsilon_{T}\right|}_{\mathrm{stat}}$ < 0.1% (‘precision’); data are recorded with a time resolution of 60 s during any individual measurement sub-run (scan) of typically 2–3 h. Note that the loop is operated in such a way that the gas composition is kept as stable as possible. Nevertheless, small drifts and sudden jumps do occur occasionally; indeed, these are unavoidable because of the concentration variations of the processed gas batches injected into the KATRIN loop. However, such variations pose no problem as long as they are monitored with the required precision, and provided that the condition ε_T_ > 0.95 is fulfilled [13]. In addition, the monitoring parameters need to be reasonably ‘true’ as well; in the case of the tritium purity, a systematic uncertainty ${\left|\Delta \varepsilon_{T}/\varepsilon_{T}\right|}_{\mathrm{sys}}$ < 3% is required [13,15]. The KATRIN-relevant requirements for the LARA monitoring are collated in Table 5 (note that in the description of uncertainties, we follow the terminology recommended in [43]).

The KATRIN LARA monitoring entails the concurrent extraction of the isotopologue concentrations, *c_x_*; the tritium purity, *ε*_T_; and the HT/DT ratio, *κ*; all are derived from the isotopologue molecule numbers, *N_x_*, which are extracted from the Raman spectra. For example, notwithstanding correction and normalization factors, the *c*_x_ values are determined using Equations (2) and (6) in Section 2 and Section 3.3, respectively. In simplified form, the *c*_x_ are described by
(8)$$c_{x}=N_{x}/\sum_{i}N_{i}\;\mathrm{using}\;N_{x}=\left[S_{x}/\left(\eta_{x}\cdot \Phi_{x}\right)\right],$$
with *S_x_* = integrated Q_1_-branch signal; *η_x_* = spectral sensitivity calibration; and Φ*_x_*= transition probability for Q_1_-branch, with the former two being experimental entities and the latter constituting a theoretical value.

Naturally, the three neutrino mass evaluation quantities in Equation (8) carry statistical and systematic uncertainties in relation to their value, $\left|\Delta X/X\right|$; these can be summarized as follows:
(i)Statistical (σ_stat_); directly associated with the determination of the *N_x_* (*S_x_*) including
-shot noise (variations of the Raman signal amplitude);-background noise (from shot noise of the fluorescence background);-readout noise (from CCD, in general negligible in our LARA measurements);-laser noise (short-term fluctuation of the *Finesse* laser, of the order <2 × 10^−4^).(ii)Systematics I (σ_cal_); associated with calibration processes, including
-uncertainty of SRM spectral intensity calibration (*η_x_*);-uncertainty from calculation of transition probabilities ($\Phi_{x}$).(iii)Systematics II (σ_ana_); associated with analysis procedures, including
-uncertainty from *ShapeFi*t, which encompasses effects from the *SCARF* background removal and other implicit analysis steps as well (*S_x_*).

Note that the two parts of the systematics are combined into a total systematic uncertainty, σ_sys_. Note also that in the treatment of uncertainty and error propagation, we closely follow the standards outlined in reference [44]; full details of our evaluation procedures are given in reference [32].

In order to demonstrate the performance of our KATRIN LARA system, some example data from the KNM1 campaign are shown in Figure 7. Particular emphasis is placed on the aspects associated with the monitoring of the key parameter, ε_T_, and the required/achieved precision and trueness of the results, as well as the related concentrations of the isotopologues T_2_, DT, and HT.

The relevant numerical values, and their uncertainties, for a selected sub-run period of about 2 h 20 min duration are collated in Table 5. By and large, the results shown in the Figure and the table confirm that the LARA monitoring fulfils all KATRIN requirements and even surpasses them. Mostly, the data speak for themselves; however, a few explanatory remarks may elucidate certain aspects of the observed monitoring data.

First, the particular tritium purity monitoring interval of two weeks (bottom data panel of Figure 7) has been chosen to encompass typical “disruptive” events in the gas circulation through the KATRIN loop. Namely, these are two “jumps” at the time when a new batch of reprocessed tritium gas enters the loop (typically on the time scale of about once per week, and marked by the (green) asterisks); and a brief interruption of the circulation for “loop maintenance” (normally very rare during a measurement campaign). However, during and after said events, the tritium purity remains well above the *ε*_T_ > 95% requirement; and the statistical uncertainty stays nearly unchanged.

Second, the expanded display of the *ε*_T_ data (for one selected sub-run period) clearly reveals that the statistical data scatter is well within the KATRIN-required statistical uncertainty, σ_ν,stat_. The fluctuation for individual data points (with 60 s measurement time each) is indicated by the red-tinted error band around the data points, yielding an uncertainty value of about σ_stat_ ≡ σ_exp_ ~ 3 × 10^−4^
≪ σ_ν,stat_ = 1 × 10^−3^ for the displayed monitoring period. The other, blue-tinted error band represents the total systematic uncertainty—or trueness—σ_sys_, as outlined further above in this section. With a value of σ_sys_ ~ 2 × 10^−3^, this is an order of magnitude better than the required σ_ν,sys_ = 3%. All relevant numerical values are collated in Table 5.

Third, the concentration data for all three radioactive isotopologues T_2_, DT, and HT are by and large dominated by signal shot noise, resulting in absolute scatter around the median of no more than 1–2 × 10^−3^. Note that as for the tritium purity parameter the statistic and systematic errors are indicated by the red- and blue-tinted error bands, respectively. It also should be noted that the relatively poor trueness for the concentration values for DT and HT is linked to the line deconvolution procedure (see Section 4.2) for the former, and to the increased uncertainty of the SRM spectral calibration at longer wavelengths (see Section 3.3) for the latter. Note as well that the three radioactive isotopologues T_2_, DT, and HT add up to 99.774% of the gas composition; the minute remainder distributes over the isotopologues H_2_, HD, and D_2_.

Overall, the data of Figure 7 and the collated values in Table 5 confirm that the monitoring of the KATRIN gas composition via laser Raman spectroscopy is very successful, and by and large surpasses the specifications. This provides a high level of confidence that, at least from the point of view of the gas supplied into the WGTS, no unwelcome surprises are expected in the evaluation of the neutrino mass, based on the analysis of the β-spectrum of tritium.

### 4.5. Remarks on Chemical and Radio-Chemical Reaction Products

When inspecting the Raman spectra of the gas mixtures investigated in this study, together with temporal evolution of the relative molecular concentrations, one observes both significant changes in the composition of the gas and the appearance of spectral features which are not associated with the Raman bands of the six hydrogen isotopologues. These can be ascribed to, in principle, well-known processes of the interaction of radioactive tritium and surface materials of the vessel—in the case of KATRIN, a range of steel alloys—in which it is contained or circulated. They include (i) the products from the actual β-decay of tritium; and (ii) the formation of secondary molecular products in reactions with hydrogen and carbon liberated from the steel wall (or deposits on it). In the following, a brief summary is given of the various reaction products observed during the three measurement campaigns.

#### 4.5.1. Products Associated with the β-decay of Tritium—Hydrogen and ^3^He Atoms

In the spectra of LOOPINO and KNM1, clear atomic line emissions are observed, in particular the Balmer-α line of hydrogen near 656 nm. This atomic emission has its origin in either of two processes, namely that molecular hydrogen isotopologues dissociate under the impact of β-electrons from the tritium decay, and/or the daughter molecule of the tritium β-decay—see Equation (1)—decomposes further according to (see, e.g., Reference [45]):^3^HeX^+^ → X + ^3^He^+^ or X^*^ + ^3^He^+^ (with X = H, D, T).(9)

It should be noted that, other than in Raman spectroscopy of molecules, the different masses of the isotopes only marginally influence the position of the (atomic) emission lines. For the hydrogen isotopes, the Balmer-α transitions for deuterium (^2^H, or D) and tritium (^3^H, or T) differ from those of protium (^1^H) by only 4.0 cm^−1^ and 5.5 cm^−1^, respectively, corresponding to 0.17 nm and 0.24 nm [46]. With the resolution of the Raman spectrometer utilized in this study (Δλ~1 nm, see Table 2), the individual isotopic lines cannot be resolved. However, a fit of the superimposed Balmer-α lines clearly reveals that it gravitates to the spectral position associated with the transition in atomic tritium. The recombination of the ^3^He^+^ ion in reaction (9) will result in the generation of neutral atoms; like for the tritium atom, these can be excited by β-electron impact to yield atomic line transitions (see, e.g., reference [47]). However, it should be noted that in the experiment described in [35], helium was significantly more abundant (several 100 mbar of added ^4^He), and much longer Raman signal acquisition times were used. Thus, under normal KNM1 operating conditions in which helium is extracted from the loop by the permeator, helium line transitions are below our light detection limits.

#### 4.5.2. HT and Tritium-Substituted Methane from Surface-Mediated Reactions

Probably the most widely studied effect of tritium interaction with the walls of containment vessels is that of surface-mediated or bulk-extraction reactions, mostly in the context of studies related to the generation of energy from nuclear fusion (see, e.g., references [48,49,50]). The two reaction products HT and tritium-substituted methane (most likely in the form of CT_4_, although mixed-substituted species might be expected as well) can be traced in the spectra displayed in Figure 4. When plotting the time evolution of all observed constituents in the gas mixture from these spectra, the accumulation of both products is evident for the LOOPINO data, while only HT and its temporal changes (see Figure 6) are visible in the spectra for KNM1. This is not surprising since, in the latter case, any methane species is removed from the circulating gas by the permeator, and thus they do not accumulate to quantities which are detectable above the noise level of our KATRIN LARA monitoring system.

The mechanisms for the generation of HT and CT_4_ in a “pure” T_2_ gas environment are by and large dominated by reaction paths associated with the extraction of hydrogen (H) and carbon (C) atoms from the steel walls of the KATRIN loop structure.

Hydrogen is available for the exchange reaction with tritium—to generate HT—via two basic routes. First, water (H_2_O) residues on the surface mediate the production of HTO and HT (see, e.g., reference [51] and references therein). The presence of traces of water in the system is expected since the loop part containing the buffer vessels as well as the long feed line to the WGTS cannot be baked out under vacuum before they are exposed to tritium, making the adsorption of H_2_O on the surfaces nearly unavoidable.

Second, equally unavoidable is the well-known presence of hydrogen “trapped” in the steel bulk during manufacture, and which can migrate through the material (see, e.g., reference [52] and references therein). In a closed system, both HT production processes should diminish over time, with the available hydrogen being consumed (as was shown in the KATRIN system during the LOOPINO campaign, see reference [41]). However, the actual KATRIN loop constitutes a regulated, “open” loop; thus, it is likely that small amounts of water might still enter into the system via the feed from the gas reservoir.

In the studies of materials for use in fusion reactors, it was found that numerous types of steels lead to a measurable amount of methanes when exposed to tritium for longer periods of time by the extraction of carbon atoms from the steel [48,49]. The same investigators, and others, also found that (at room temperature) the exposure of steels to deuterium does not result in the generation of methane. Note, however, that for elevated temperatures, decarburization of steel in a hydrogen atmosphere readily occurs (see, e.g., reference [53]).

Clearly, this shows that the isotopes of hydrogen do not behave identically. The radioactive ß-decay process of tritium seems to constitute a necessary prerequisite for the formation of methane, in simplified form according to C + 2·T_2_ + 1 eV → CT_4_ (a more detailed proposal for the reaction mechanism may be found in reference [49]).

Finally, it is noteworthy that the spectral feature observed in the range 675–680 nm in the LOOPINO spectra (Figure 4a) can be associated with β-induced fluorescence from an accumulating molecular species. This feature persists even without laser irradiation while all Raman peaks disappear; therefore, this signal is not discussed any further but is subject to a future publication.

#### 4.5.3. Reactions Observed during the FT Campaign

As stated earlier, the gas mixture circulating within the KATRIN loop was lean in tritium, with only ~1.2% of DT and <0.1% of T_2_. Thus, all tritium-mediated reaction products are expected to be present at much lower concentrations. Indeed, neither the Balmer-α line of tritium nor any ^3^He line are visible in the spectrum of Figure 4, meaning that both products are below the detection limit of KATRIN LARA system. Equally, the exchange reaction between tritium and hydrogen and carbon at the vessel surfaces is too feeble to generate sufficient HT or CT_4_ to be detectable by our KATRIN LARA system.

Finally, it should be noted that, instead of the T ⇆ H exchange mediated by the (most likely) small amounts of H_2_O at the wall surfaces, D ⇆ H exchange is observed in the circulating gas mixture during the FT campaign (for the initial gas composition see Table 3). When plotting the time evolution for D_2_ and HD (not shown here), we found a similar evolution behavior as that observed in KNM1 for T_2_ and HT, albeit with substantially different time constants: more HD was generated over time than was present in the reservoir gas fed into the loop.

## 5. Conclusions

The results presented here clearly demonstrate the capabilities and reliability of our laser Raman system for the long-term Raman spectral monitoring of the gas mixtures circulating in the KATRIN loop. Routinely, data precision of the order $\left|\Delta X/X\right|$ < 10^−3^ has been achieved, fulfilling or surpassing the monitoring requirements for the precise determination of the neutrino mass in KATRIN. A few additional remarks are worthwhile.

First, besides the monitoring capabilities for the isotopologue concentrations and the tritium purity, the spectroscopic measurements and their evaluation revealed a range of reaction-dynamic phenomena, which, on one hand, were not surprising, but which, on the other hand, still lack full theoretical modeling. In particular, the generation of HT and CT_4_ (and potentially H-/D-substituted isotopologues)—well-known reaction products when steel surfaces are exposed to tritium—require the development of a full model which incorporates all individual system segments, exhibiting vastly varying pressure, flow rate, and temperature conditions. Attempts to develop a realistic model for this are under way.

Second, the spectral identification of tritiated methanes was not unambiguous due to their rather low overall concentration in the loop (e.g., in LOOPINO < 1% after more than 50 days of circulation) and some disagreement between observed spectral features and published data. Spectroscopy experiments with “pure” CT_4_ are planned for the near future to elucidate potential discrepancies in the observed spectra, and thus to allow for a better understanding of the radio-chemical reactions in the KATRIN loop.

Finally, we would like to point out that the KATRIN LARA system, beyond the monitoring aspect being key for this investigation, could be utilized in other applications as an analytical tool for circulating gas mixtures, potentially being suitable for rapid process control. In this context, closely inspecting the spectral data utilized in Figure 4 and Figure 5, one arrives at a relative detection limit of ~8.8 × 10^−4^ (≡ 3 × σ_noise_), which equates to a lower limit of detection of LOD ≈ 0.18 mbar, for a single spectrum with 60 s acquisition time (assuming a transition line strength equal to that of the hydrogen isotopologues).

## Figures and Tables

**Figure 1 sensors-20-04827-f001:**
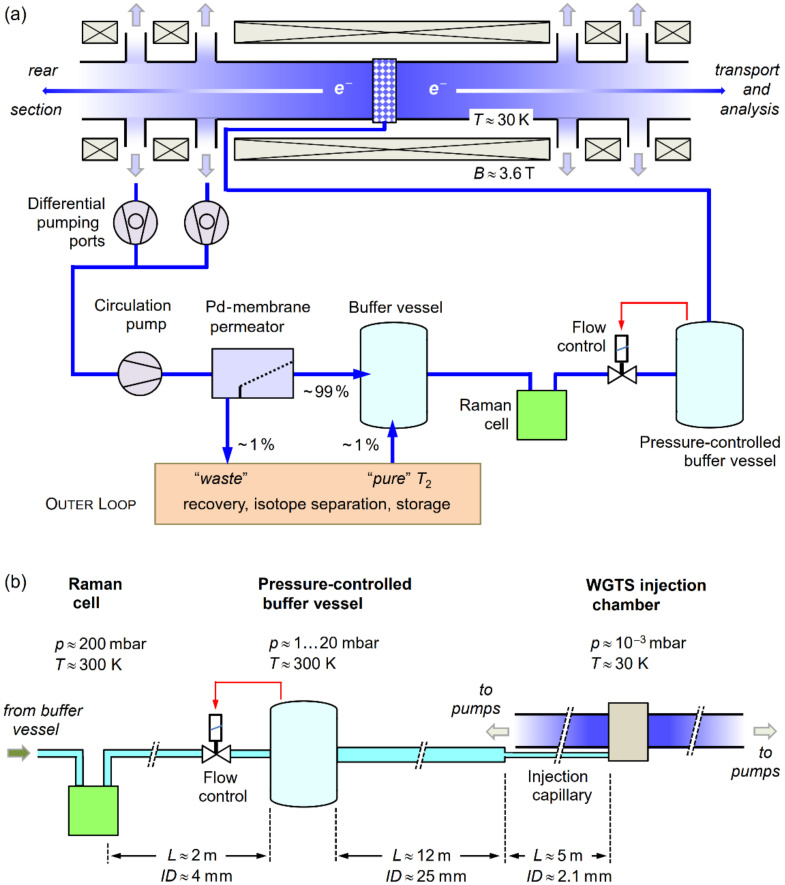
Concept of the gas circulation through the WGTS. (**a**) Schematic simplified diagram of the inner loop, incorporating the WGTS component group; note that, for clarity, not all differential pump port connections to the loop are drawn. The link to the outer loop of the TLK infrastructure for recovery, isotope separation, and storage is included conceptually. (**b**) Selected technical details of the loop segment from the Raman cell to the WGTS injection chamber; relevant numerical values for dimensions—*L* and *ID* (=inner diameter)—and operating parameters—*p* and *T—*are indicated.

**Figure 2 sensors-20-04827-f002:**
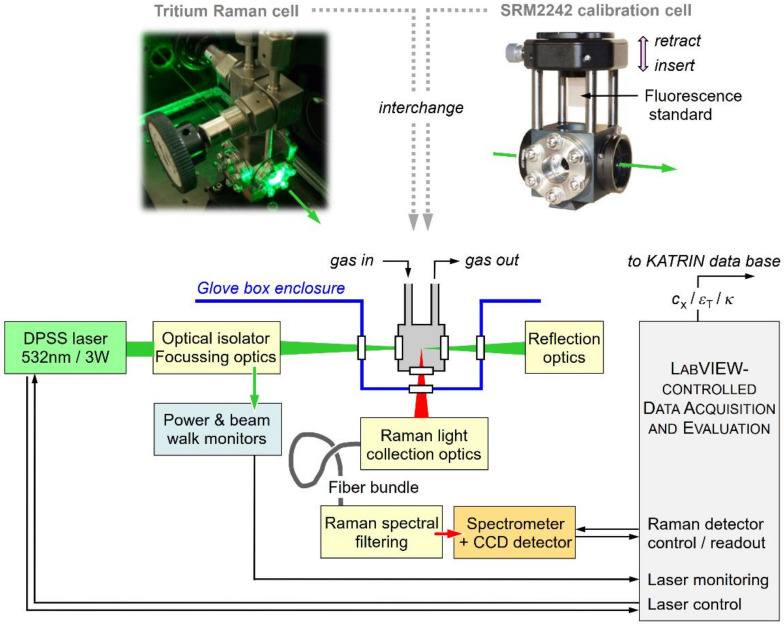
Concept of the laser Raman (LARA) monitoring setup. The LARA cell is positioned within the secondary glove box enclosure (for tritium safety); it is accessible for laser excitation and Raman light collection through anti-reflection (AR) coated windows. For in situ absolute spectral sensitivity calibration, the LARA cell can be replaced with a fluorescence standard assembly. For further details, see text.

**Figure 3 sensors-20-04827-f003:**
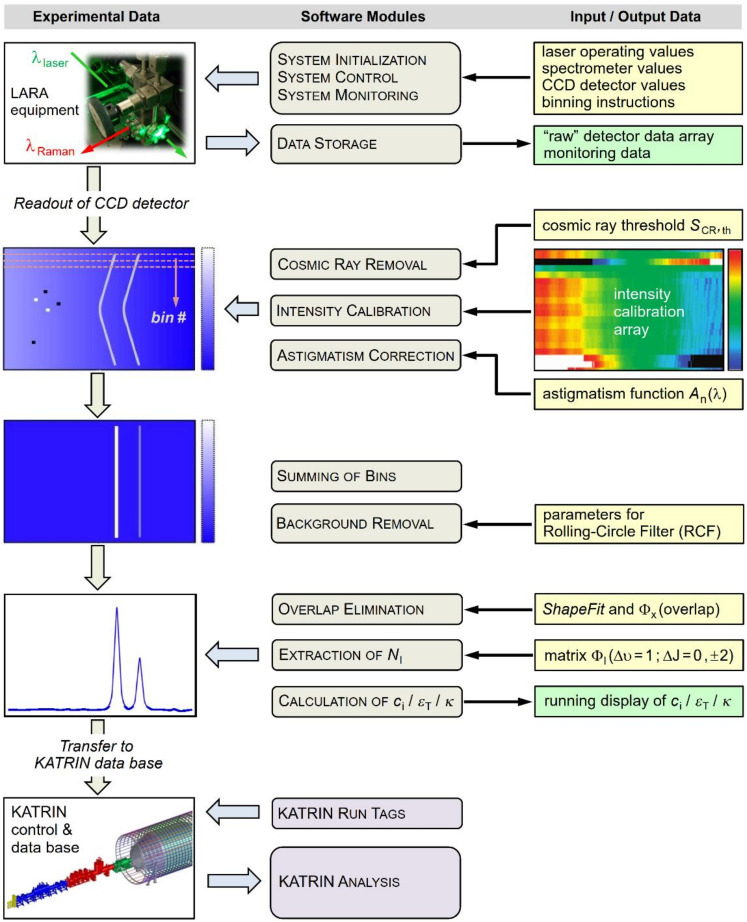
Concept of LARA system control and Raman data processing, based on the bespoke, modular *LabVIEW* suite *LARAsoft*. Note that the KATRIN control and data storage routines are not part of *LARAsoft.* For further details, see text.

**Figure 4 sensors-20-04827-f004:**
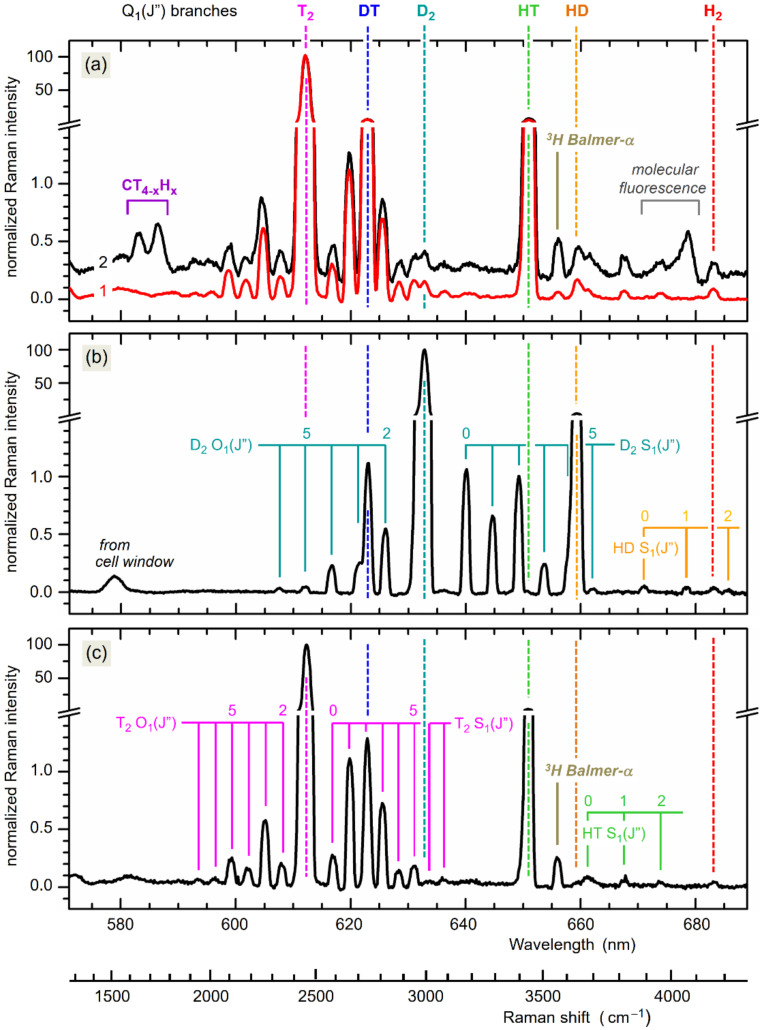
Spectra from selected pre-KATRIN and KATRIN gas circulation campaigns. (**a**) LOOPINO—12/2014: Raman spectra from the first day of circulation (trace 1) and after 55 days of circulation (trace 2, offset by +0.2); the two spectra are offset to each other for clarity. (**b**) First tritium (FT)—05/2018: Raman spectrum of the gas mixture after two days of circulation. (**c**) KATRIN ramp-up (KNM1)—03/2019: Raman spectrum recorded after five days of circulation, during the second ramp-up step. For all spectra, relevant features are annotated; for further detail, see text. Note the split intensity scales to visualize both strong and weak spectral features.

**Figure 5 sensors-20-04827-f005:**
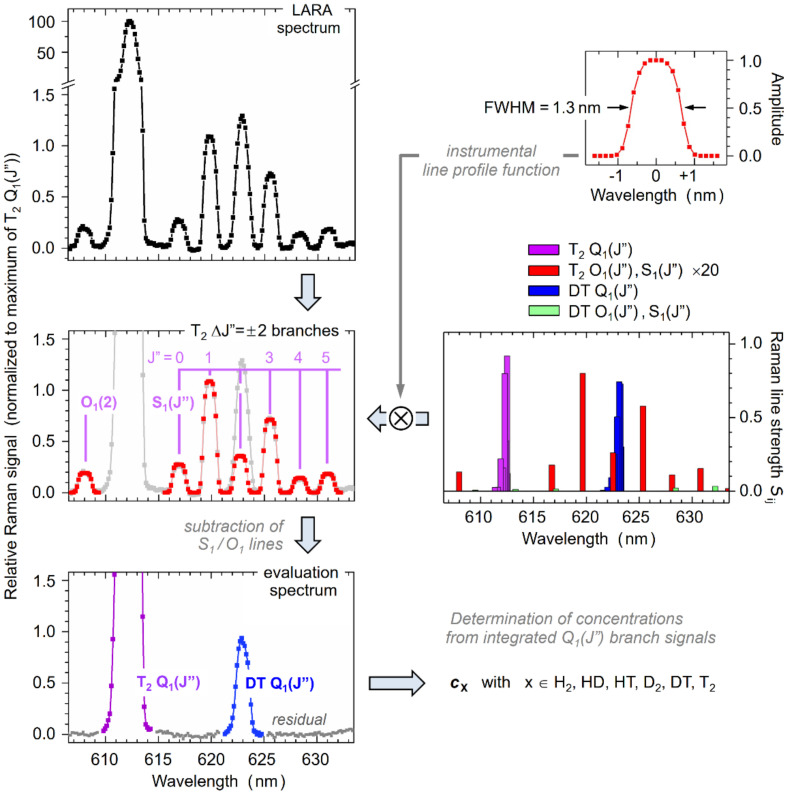
Deconvolution of spectral overlapping, exemplified for the determination of exact T_2_/DT concentrations in the KNM1 gas mixture. Top-left—segment of the spectrum in Figure 3c (the scale is split, to visualize all spectral features; center-left—theoretical O_1_(*J*″)/S_1_(*J*″) lines for T_2_, overlaid on the experimental spectrum, with the individual Raman lines annotated; bottom-left—spectrum with the O_1_(*J*″)/S_1_(*J*″) lines subtracted, leaving only the Q_1_(J″)-branches to evaluate species concentrations. Top-right—experimental line profile function, derived using *ShapeFit*; center-right—theoretical Raman line strengths φij for the O_1_(*J*″)/Q_1_(*J*″)/S_1_(*J*″) lines of T_2_ and DT, calculated for a gas temperature of T = 298 ± 2 K. Note that the values for the O_1_(*J*″)/S_1_(*J*″) lines of T_2_ are scaled by ×20, to emphasize the overlap between T_2_ S_1_(2) and DT Q_1_(*J*”).

**Figure 6 sensors-20-04827-f006:**
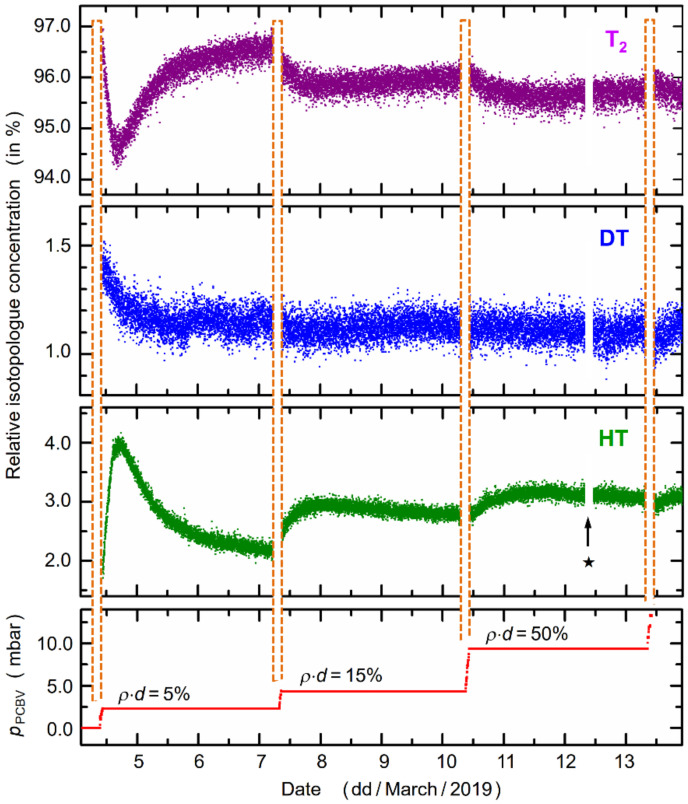
Temporal change of relative isotopologue concentrations of the tritium gas mixture circulating through the WGTS, during the first ramp-up to full KATRIN operation during 4–15 March 2019. Top panels: relative concentrations for the three radioactive isotopologues T_2_, DT, and HT. During the time period indicated by the symbol ★, no data were available (caused by an intermittent data transfer failure); during the ramp-up change periods (indicated by the dashed vertical lines), the system response does not tend to equilibrium—these data are excluded from the long-term analysis. Bottom panel: monitoring data for the pressure-controlled buffer vessel (PC-BV) during the ramp-up—the related, nominal *ρd* values for the WGTS are indicated. For further details, see text.

**Figure 7 sensors-20-04827-f007:**
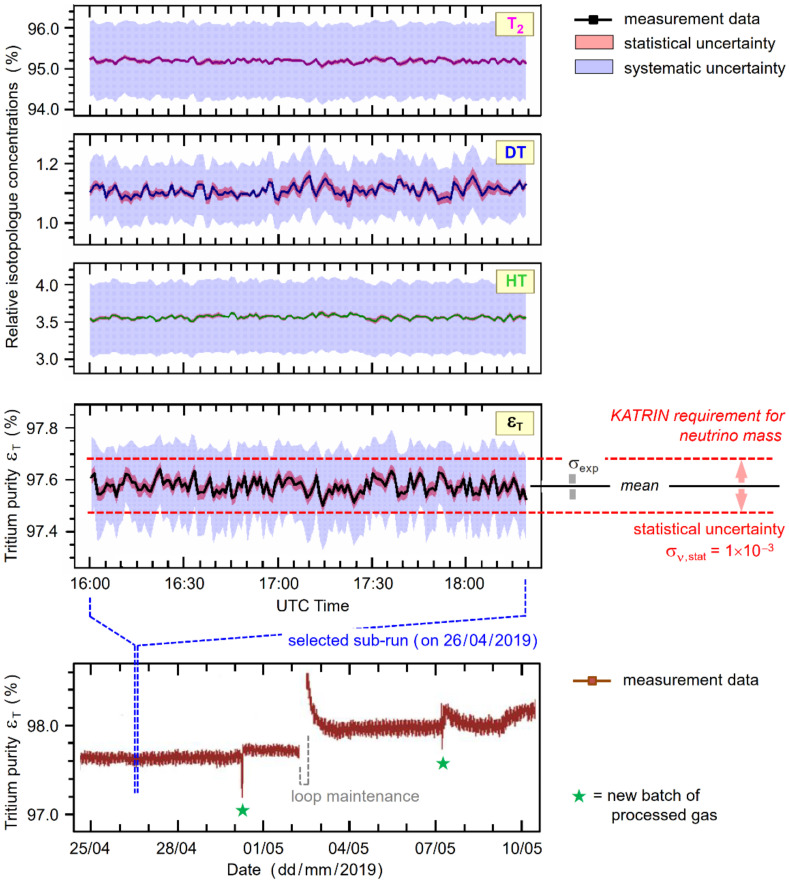
Selected measurement data from KATRIN LARA during KNM1 (April—May 2019). Top panels—concentrations of the three tritiated isotopologues T_2_, DT, and HT (in percent of the sum of all molecular gas components); the statistical and systematic uncertainties are indicated by the colored bands. Bottom panels—derived tritium purity ε_T_ for a two-week period during KNM1 (lower data trace) and an individual sub-run (upper data trace); for the latter the statistical and systematic uncertainties are indicated by the colored bands. The statistical uncertainty limit, required for the KATRIN neutrino mass calculation, is indicated on the right. For further details, see text.

**Table 1 sensors-20-04827-t001:** Equipment modules used in the KATRIN LARA setup, and their specifications.

Device	Model	Manufacturer	Specifications
Laser	Finesse	Laser Quantum, UK	Nd:YVO_4_ 2^nd^ harmonic (TEM_00_); λ_L_ = 532 nm; P_L_ = 5 W (CW)
Raman filter	RazorEdge LP03-532RU	Semrock, USA	T > 0.97 for λ_Raman_ > 537_._nm; T < 10^−6^ for λ_L_ = 532 nm
Fiber bundle	Custom “slit-to-slit”	CeramOptec, Germany	48 individual fibers, core = 100 μm; Bundle height = 6 mm
Spectrometer	HTS	PI Acton, USA	f = 85 mm, with f/# = 1.8; λ_range_ = 500–750 nm (fixed); Δλ ≈ 1 nm (for slit = 100 μm)
CCD array detector	Pixis 2KB	Princeton Instruments, USA	Back illuminated; 2048 × 512 pixel (27.6 × 6.9 mm); T_CCD-chip_ ≤ −70 °C; Dark noise ≈ 10^−3^ e^−^·s^−1^·pixel^−1^
Acquisition software	LARAsoft	TLK in-house	Device control; Data acquisition; Spectrum analysis (written in LabVIEW)

**Table 2 sensors-20-04827-t002:** Settings of key operating parameters for (i) LOOPINO-3—loop test measurements (without WGTS; unpublished results); (ii) FT—first tritium campaign, with reduced tritium content [21]; and (iii) KNM1—first KATRIN neutrino mass data taking [22].

Parameter/Setting		Units	LOOPINO	FT	KNM1
Pressure in LARA Cell, *p*_RC_		mbar	149	190	190
Tritium content in the gas mixture, *ε*_T_			~0.93	~5 × 10^−3^	>0.97
Column density, *ρd* (fraction of nominal *ρd*_max_)		%	n/a	~100	25
Laser power, *P*_L_		W	1.5	4.0	3.0
Spectral line resolution (FWHM),	Δλ	nm	1.15	1.15	1.15
	$\Delta \tilde{\nu }$	cm^−1^	28.2	28.2	28.2
CCD acquisition time (single spectrum), *t*_SS_		s	58.5	58.5	58.5

**Table 3 sensors-20-04827-t003:** Nominal hydrogen isotopologue concentrations (in percent) of the reservoir gas supplies used during the different measurement campaigns.

*c*_x_ for	T_2_	DT	D_2_	HT	HD	H_2_
LOOPINO ^(1)^	89.4	5.4	<0.1	5.0	<0.1	<0.1
FT ^(2)^	<0.1	1.2	97.6	<0.1	0.9	<0.1
KNM1 ^(2)^	96.8	1.6	<0.1	1.2	<0.1	0.1

^(1)^ Pre-transfer analysis using gas chromatography (GC). ^(2)^ Pre-transfer analysis using dedicated laser Raman system.

**Table 4 sensors-20-04827-t004:** Comparison of the Raman line strengths Sij, and the experimental and theoretical signal amplitudes—A_RS,theory_ and A_RS,exp_, respectively—for the T_2_ S_1_(*J*”) Raman lines of Figure 5.

T_2_ Raman Line	S_1_(0)	S_1_(1)	S_1_(2)	S_1_(3)	S_1_(4)	S_1_(5)
S_ij_ ^(1)^	0.00889	0.03990	0.01302	0.02841	0.00540	0.00737
A_RS,theory_ ^(2)^	0.241	1.094	0.355	0.785	0.151	0.209
A_RS,exp_	0.253	1.094	---	0.770	0.154	0.200

^(1)^ Raman line strengths S_ij_ based on the transition probability function $\Phi \left(\Delta \upsilon ;J”\right)$, calculated by LeRoy [33]. ^(2)^ The theoretical signal amplitudes, derived from Sij, are normalized to the S_1_(1) line amplitude.

**Table 5 sensors-20-04827-t005:** Typical monitoring parameters for the gas composition in KATRIN: comparison of the specification requirements and the values achieved during first KATRIN neutrino mass data taking. Precision and trueness are given relative to their value, $\left|\Delta X/X\right|$.

		KATRIN Requirements ^(1)^			Achieved during KNM1 ^(2)^		
Parameter		Value	Precision	Trueness	Value	Precision	Trueness
Concentrations, *c_x_*	for T_2_				0.95193	4.7 × 10^−4^	9.7 × 10^−3^
	for DT				0.01109	1.5 × 10^−2^	9.2 × 10^−2^
	for HT				0.03562	6.4 × 10^−3^	1.3 × 10^−1^
Tritium purity, *ε*_T_		>0.95	1 × 10^−3^	3 × 10^−2^	0.97576	2.8 × 10^−4^	1.6 × 10^−3^
Ratio of impurities HT/DT, *κ*		---	---	10 × 10^−2^	3.212	2.5 × 10^−2^	3.3 × 10^−2^

^(1)^ Originally defined in reference [13], amended in reference [15]. ^(2)^ From the “snapshot” sub-run data acquired on 26 April 2019 (see Figure 7).

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
