# Peer review of "Quantitative Long-Term Monitoring of the Circulating Gases in the KATRIN Experiment Using Raman Spectroscopy"

_sensors, 2020, doi:10.3390/s20174827_

Round 1

Reviewer 1 Report

The paper nicely presents the details of the monitoring system. The idea of the usage of spectroscopic measurements for monitoring the gas system has been presented well. 

Author Response

We would like to thank the reviewer for his efforts in reading this long manuscript. We are pleased, that we seem to have written the manuscript in a sufficiently clear way for the reviewer to recommend its publication “as is”.

Reviewer 2 Report

The manuscript describes a dedicated Raman setup for the monitoring of tritium and its isotopologues in the KATRIN experiment (measurement of neutrino mass). It is well written and provides a detailed and comprehensible body of work. The interplay of the Raman measurement and its importance to the whole experiment are well documented and the results are clearly presented.

I have only minor suggestions:

In section 3.2 the Raman setup is described extensively, but there are some details missing. This regards the gas cell, where the laser is coupled in, then back-reflected for improved laser power. There is however no information about the optical system (lenses etc.) as well as information about beam size and shape at the location of the measurement. Similarly, the collection optic is not described sufficiently, as it would be interesting which kind of optics are used to couple the scattered light into the collection fiber. In the manuscript, the authors write that a slit-to-slit fiber is used. When classical lenses are used for focusing, one would expect losses due to vignetting. Most often round-to-slit fibers are used for this reason. Can the authors elaborate on these points?

Line 790: “…around the median of nor more…” There is a typo, it should probably say “of no more”

Line 879: The “there” should probably be a “therefore”

Author Response

We would like to thank the reviewer for his careful reading of the text. In particular, we appreciate the suggestions he made respective the description of the LARA system in Section 3.2.

Personally we think that by and large the text is sufficient as it stands; for one, a full, detailed description of all individual components would add substantially to the already lengthy manuscript, and second it would replicate information which is available from several detailed publications. However, we understand the frustration of the reviewer that for possibly interesting details he needed to consult other publications. In order to alleviate the problem for other readers, we have amended the text with a few short sentences, addressing relevant points in the reviewer’s comment.

+++++

Reply respective Section 3.2 (broken down into the the individual comment points made by the reviewer, and amended at the relevant locations in the text):

  • beam size and shape at the location of the measurement [original lines 300-301; in revision lines 301-305] …gas cell by a long focal-length lens, generating an elongated, near-cylindrical interaction volume of about 6 mm in length and a beam waist of ~100 m In variation to the original design now the laser beam is back-reflected through the cell, using a combination of a lens (with the same focal lens as utilized for the input coupling) and a dielectric mirror. This replicates the primary excitation volume, effectively doubling …
  • Collection optics [inserted in original line 307; in revision lines 311-313]  … fiber bundle (for its dimension see Table 1). Note that a set of two plano-convex lenses is used to provide 1:1 imaging of the Raman interaction volume onto the entrance of the fiber bundle, with an acceptance angle matched to the numerical aperture of the fiber. Prior to entering …
  • Slit-to-slit fiber bundle. Here only a note to the reviewer: With reference to Figure 2, the light from the excitation volume in the cell is collected at 90°; thus the cylinder cross section is most efficiently imaged / focussed onto a fiber “slit” with the same dimension. The lenses have diameters large enough to by and large avoid any vignetting. This point has partially been addressed in the wording of the “collection optics” amendment.

+++++

Typo in line 790: [in revision line 819]  Corrected to: … of no more ….

+++++

Typo in line 879: [in revision line 903]  Corrected.to:  … therefore,

Reviewer 3 Report

I congratulate the authors on what is an absolutely outstanding piece of analytical science. This is a comprehensive and extremely well-written piece of work. The system is beautifully described, the figures are first-class and the attention to detail in the analysis is exceptional.
I have zero hesitation in recommending acceptance for publication in Sensors.

My only quibble is that at key points in the paper the authors have an irritating habit of telling the reader to go and look elsewhere for the interesting details. A few examples are:
Line 195 I wanted to know how well the system had performed so I had to go and look-up ref [22]. A simple statement (one sentence!) of the results would have been nice.
Line 363 To determine the system's response function is non-trivial. A short paragraph summarising how this was done, rather than just the reference would be useful to the reader.
Line 380 A table showing the comparison of observed and calculated intensities for a half-a-dozen relevant lines would be interesting.
Line 402 "rather promising" is a tease! A small table showing the comparison would be more convincing.

I emphasise that these are only suggestions to make the paper even better, they are not requested revisions. Again, my congratulations to the KATRIN collaboration.

Author Response

In the first instance, we like to thank the reviewer for his diligent review. Although the reviewer has indicated that his suggestions are not meant to be requested revisions, we agree that he raises some valid points. Thus we have augmented our text respective most of the suggestions, as outlined below.

+++++

Reply to line 195 query:   Although the paper addresses the Raman measurements during KATRIN, and thus neutrino mass data are not of direct relevance to the Raman monitoring, for the “curious” we have added the actual numerical value.

[added at the end of line 195]:  … now been published, yielding an upper bound for the electron neutrino mass of ?() < 1.1 eV/c2 (90% confidence limit) [22].

+++++

Reply to line 363 query:   The procedure is actually “sketched” earlier in the paragraph, with reference to Figure 2. However, to clarify this we have added a couple of procedural sentences:

[in revision lines 369-373]:… in Figure 2). In brief, the SRM-standard is lowered into the laser beam path so that fluorescence is generated at exactly the same position as the ordinary Raman excitation in the gas. The fluorescence from this same-size volume passes through the Raman light collection system. The recorded fluorescence spectrum is then compared to the NIST SRM2242 reference data, thus providing an absolute intensity calibration including the full optical detection system. Details …

+++++

Reply to line 380 query:   Here a comparison for the line intensities can solely be given for O1- and S1-branch lines; only these can be resolved with the spectrometer used in the KATRIN experiment. For the Q1-branch lines higher resolution is required, as used during the depolarization experiments described in ref [35]. The good agreement theoretical vs experimental line intensities (proportional to transition probabilities) can be gleaned from Figure 5. There the subtraction of experimental and theoretical line profiles results in near-zero residuals (the theoretical amplitudes are deduced using equation 5). We appreciate the reviewers desire to see actual numerical values; however, a full absolute-value comparison would require lengthy explanations and describe secondary calculation steps. A relative comparison respective a reference line is the only way to provide such information in concise form. We have done this now for the T2 S1(J”) lines shown in Figure 5, collating the results in a new Table 4.

Accordingly, we have amended line 380 [in the revision lines 393-394] with

… has been applied; example data are collated in Table 4, referring to the Raman transition lines shown in Figure 5 in Section 4.3 below.

Said new Table 4 has been inserted in close proximity to Figure 5. To refer to it we have added [at the end of line 611, in the revision line 629]: 

… equation (5). Related numerical data are collated in Table 4.

As a consequence, the former Table 4 on page 21 has been renumbered as Table 5; and reference to this table has also been corrected in the associated text.

+++++

Reply to line 402 query:   The research related to TriHyDe is currently being written up for a PhD thesis. The results reported in [38] were for non-tritium isotopologues (the certification for tritium operation was not yet complete at that date). The measurements with tritium-containing isotopologues have just been completed and will be reported in said PhD thesis later this year, and in an associated journal publication.

We wish not to preempt the thesis results, but in order to improve on the “teasing” we have amended the text in the following way: 

[in revision lines 403-406]:   … This has been done for the non-radioactive isotopologues of hydrogen – H2, HD and D2 – in a measure­ment setup at TLK, dubbed “HyDe” [36,37]. In brief, for the calibration of LARA data against known concentration data, two combined datasets are necessary, namely (i) a binary data set with only the initial isotopologues not in thermal equilibrium; and (ii) a tertiary data set with all three corresponding isotopologues in thermal equilibrium. The agreement between the two methods was excellent, mostly <3 % for the Q1-branch signals.

[in revision lines 416-421]:   … A tritium-compatible mixing system has now been con­structed – conveniently named “TriHyDe” – with which all six hydro­gen isotopologues can be tack­led [38]. In spite of the aforementioned complications first results look rather promising. Foremost, the HyDe calibration for H2:D2 mixtures could be replicated, yielding agreement to within 2-3%. Measurements of binary and tertiary mixtures containing tritium are now almost complete, and evaluation of the data is ongoing. Just as an example, for the binary mixture D2:T2 the calibration factors are accurate to less than 2%. The description of the related TriHyDe system setup and results from the full calibration sets for all six isotopologues will be subject to a forthcoming publication.